# *Toxoplasma gondii* infections are associated with costly boldness toward felids in a wild host

Eben Gering [1,2,11], Zachary M. Laubach [1,3,4,10,11 ✉], Patty Sue D. Weber[5], Gisela Soboll Hussey[6], Kenna D. S. Lehmann [1,4], Tracy M. Montgomery[1,4,7], Julie W. Turner [1,4,8], Wei Perng[9], Malit O. Pioon[4], Kay E. Holekamp [1,4] & Thomas Getty [1]

*Toxoplasma gondii* is hypothesized to manipulate the behavior of warm-blooded hosts to promote trophic transmission into the parasite's definitive feline hosts. A key prediction of this hypothesis is that *T. gondii* infections of non-feline hosts are associated with costly behavior toward *T. gondii*'s definitive hosts; however, this effect has not been documented in any of the parasite's diverse wild hosts during naturally occurring interactions with felines. Here, three decades of field observations reveal that *T. gondii*-infected hyena cubs approach lions more closely than uninfected peers and have higher rates of lion mortality. We discuss these results in light of 1) the possibility that hyena boldness represents an extended phenotype of the parasite, and 2) alternative scenarios in which *T. gondii* has not undergone selection to manipulate behavior in host hyenas. Both cases remain plausible and have important ramifications for *T. gondii*'s impacts on host behavior and fitness in the wild.

[1] Michigan State University, Department of Integrative Biology and Program in Ecology, Evolution and Behavior, East Lansing, MI, USA. [2] Nova Southeastern University, Department of Biological Sciences, Halmos College of Natural Sciences and Oceanography, Fort Lauderdale, FL, USA. [3] University of Colorado Boulder, Department of Ecology and Evolutionary Biology, Boulder, CO, USA. [4] Mara Hyena Project, Narok County, Kenya. [5] Michigan State University, Department of Large Animal Clinical Sciences, College of Veterinary Medicine, East Lansing, MI, USA. [6] Michigan State University, Department of Pathobiology and Diagnostic Investigation, College of Veterinary Medicine, East Lansing, MI, USA. [7] Max Planck Institute of Animal Behavior, Department for the Ecology of Animal Societies, Konstanz, Germany. [8] Memorial University of Newfoundland, Department of Biology, St. John's, NL, Canada. [9] LEAD Center & University of Colorado, School of Public Health, Aurora, CO, United States. [10] Present address: Department of Ecology and Evolutionary Biology, University of Colorado Boulder, Boulder, CO, USA. [11] These authors contributed equally: Eben Gering, Zachary M. Laubach. ✉email: zachary.laubach@colorado.edu

*T*oxoplasma gondii provides an infamous example of putative host-manipulation by a parasite[1]. Parasite transmission can occur by ingestion of *T. gondii* oocysts shed from felids (the definitive host), consumption of infected tissue from intermediate hosts, and congenital infection[2,3]. Across this protist's diverse array of warm-blooded hosts, infections are linked to reduced avoidance of, or even attraction to, the odor of feline urine[4–6]. The parasite's hypothesized ability to alter host responses to indirect cues of feline presence is thought to have evolved by natural selection on the parasite to increase trophic (prey to predator) transmission. This could benefit *T. gondii* since the parasite undergoes sexual reproduction within definitive feline hosts to produce recombinant, environmentally stable propagules called oocysts[7,8]. *T. gondii* also induces other potentially manipulative behaviors in intermediate hosts, including behavioral boldness[9]. If this boldness results in lethal contact with felines, it could similarly promote trophic transmission of *T. gondii* at the expense of intermediate hosts' fitness in nature[10].

While *T. gondii* is among the best-studied putative host manipulators, and also causes substantial disease burden in human hosts[7,11], its effects on host behavior have overwhelmingly been studied in laboratory animals and humans. A smaller body of research from nature, where *T. gondii* co-evolves with intermediate and definitive hosts, suggests that infection-related behavior might decrease host fitness[12,13]. For example, wild-caught rodents harboring naturally occurring infections exhibit reduced avoidance of odor cues from local felids[14], as well as elevated activity[15], reduced neophobia[16], and higher rates of capture in human traps in captive and semi-captive settings[16]. In wild sea otters, infections are also associated with both neuropathy and shark predation[17]. Yet, no prior study has examined the relationship between *T. gondii* infections in wild hosts and naturally occurring interactions involving the parasite's definitive felid hosts.

In this study, we adapt the spotted hyena system to better understand the links between *T. gondii* infection and fitness-related behavior in free-living hosts towards felids (While the design and objectives for this body of work were presented along with preliminary findings in a non-peer-reviewed *Festschrift* volume[18], the present manuscript includes more rigorous models that include additional candidate covariates to arrive at somewhat modified conclusions. We also present novel syntheses of our research findings in light of existing literature and point to important next steps in studying putative parasitic manipulation in wild hosts). More specifically, we used blood samples and detailed field observations spanning three decades to accomplish three goals: 1) identify demographic, social, and ecological determinants of *T. gondii* infection in the spotted hyena (*Crocuta crocuta*); 2) test whether *T. gondii*-infected hyenas exhibit greater behavioral boldness in the presence of lions, and 3) test whether *T. gondii*-infected hyenas have higher rates of lion-inflicted mortality. Our data were collected in a natural setting in Kenya where hyenas frequently interact with lions (*Panthera leo*), which are not only definitive *T. gondii* hosts[11] but are also the leading cause of hyena injuries and mortality[19,20]. Despite clear risk, hyenas engage with lions to defend territories, protect relatives, and/or compete for food. Tension between the benefits and costs of these interactions likely explain findings of stabilizing selection on hyena boldness toward lions, favoring individuals with intermediate phenotypes[21]. This study system permits us to characterize relationships among *T. gondii* infection and naturally occurring behaviors that have fitness consequences.

Here, we show that wild hyena cubs infected with the parasite *T. gondii* exhibit costly behavioral boldness when interacting with lions and that infected cubs experience a higher probability of lion mortality than their uninfected group mates. Furthermore, our results indicate that *T. gondii* infection prevalence is age structured with older animals being more likely to be infected.

## Results

One hundred and eight (108) of 166 surveyed hyenas (65%) had IgG antibodies to *T. gondii*, indicating prior exposure to the parasite. Thirty-seven individuals (22%) tested negative, and 21 hyenas (13%) yielded results within the "doubtful" range of the assay (Supplementary Fig. 1). In keeping with prior studies[22], we combined individuals with negative and doubtful diagnoses into a single category, treating them as uninfected in the analysis. A subset of 60 plasma samples were also tested using IFAT diagnostics to confirm consistency between ELISA and IFAT (Supplementary Fig. 2; Pearson's r(58) = 0.70, $p < 0.001$).

Table 1 shows the distribution of *T. gondii* infection prevalence across demographic, social and ecological variables (sex, age, dominance rank, and livestock density). We observed no differences in infection prevalence between male vs. female hyenas (61% vs. 68%; $P = 0.40$). Hyena cubs (35% infected) had lower infection prevalence than subadults (71%) and adults (80%; overall $P$-difference <0.001). Dominance rank was not associated with the probability of being infected ($P = 0.95$). Hyenas sampled in areas of high livestock density did not differ in their infection prevalence (76% vs. 62%, $P = 0.15$) compared to hyenas from areas of low livestock density.

Adjusting for potential confounding variables did not change findings (Table 1). Hyena sex, dominance rank, and living in high vs. low livestock density areas were not associated with infection status (odds ratio [OR] for male vs. female hyenas: 0.70 [95% CI: 0.33, 1.47] OR for standardized dominance rank 0.95 [95% CI: 0.43, 2.07]), and OR for high vs. low livestock density 0.56 [95% CI: 0.20, 1.46]). *T. gondii* was more prevalent in older individuals (OR for subadults vs. cubs: 5.05 [95% CI: 1.80, 15.17]; OR for adults vs. cubs: 8.11 [95% CI: 3.59, 19.32]).

Second, we investigated associations of *T. gondii* infection with boldness toward lions, as indicated by minimum approach distance to lion(s). Supplementary Table 1 shows bivariate associations between a) hyenas' minimum approach distances toward lions, and b) candidate confounding variables of the relationship between *T. gondii* infection and hyena approach distance from lions. At alpha = 0.05, shorter minimum approach distances were seen in female and older hyenas (i.e., subadult or adult), among higher dominance rank hyenas and in areas with high livestock density.

Given the age structure of *T. gondii* prevalence in hyenas, coupled with our observations that older hyenas consistently approach lions more closely than cubs, we conducted separate analyses of behavioral covariates of infection in cubs vs. older individuals. Here we report all estimates as square root transformed distances in meters from lions, unless otherwise noted. Among cubs, infected individuals had a shorter minimum approach distance from lions (−3.19 [95% CI: −5.57, −0.81]) than their uninfected counterparts after controlling for sex and age in months at the time of interaction (Fig. 1A, Table 2). Among subadults and adults, infection was not related to minimum approach distance (0.27 [95% CI: −0.19, 0.72]; Fig. 1B, Table 2). Limiting our dataset to hyena-lion interactions recorded after the diagnosis dates for seropositive individuals and prior to the diagnosis dates of seronegative individuals, we observed no association (0.48 [95% CI: −0.58, 1.53] between infection status and boldness behaviors. Similarly, excluding subadult and adult hyenas with a "doubtful" *T. gondii* diagnosis did not materially change the results (0.02 [95% CI: −0.87, 0.92]) related to minimum approach distance between infected vs. uninfected hyenas.

In sensitivity analyses, we assessed the potential effects of known determinants of hyena boldness behaviors in this

**Table 1 Prevalence of *T. gondii* infection among 166 spotted hyenas from the Masai Mara, Kenya, and its relationship to demographic, social, and ecological variables.**

| | % (N) | | OR (95% CI) infected vs. uninfected | |
|---|---|---|---|---|
| | **Uninfected** | **Infected** | **Unadjusted[a]** | **Adjusted[b]** |
| | *n* = 58 | *n* = 108 | | |
| Sex | | | | |
| Female | 32% (31) | 68% (65) | 1.00 (Reference) | 1.00 (Reference) |
| Male | 39% (27) | 61% (43) | 0.76 (0.40, 1.45) | 0.70 (0.33, 1.47) |
| Age at diagnosis[c] | | | | |
| Cub (<12 mos) | 65% (32) | 35% (17) | 1.00 (Reference) | 1.00 (Reference) |
| Subadults (12−24 mos) | 29% (10) | 71% (25) | 4.71 (1.88, 12.49)[f] | 5.05 (1.80, 15.17)[f] |
| Adult (>24 mos) | 20% (16) | 80% (66) | 7.76 (3.55, 17.80)[f] | 8.11 (3.59, 19.32)[f] |
| Dominance Rank[d] | | | | |
| Standardized rank (−1: 1) | 42% (40) | 58% (56) | 1.02 (0.53, 1.96) | 0.95 (0.43, 2.07) |
| Livestock density[e] | | | | |
| High | 24% (8) | 76% (25) | 1.00 (Reference) | 1.00 (Reference) |
| Low | 38% (50) | 62% (83) | 0.53 (0.21, 1.22) | 0.56 (0.20, 1.46) |

[a]From a logistic regression model where the explanatory variable of interest is each socioecological characteristic, and the outcome is infection (yes. vs. no).
[b]Adjusted models control for a hyena's sex, age at diagnosis, and livestock density.
[c]Age was assessed on the date the hyena was diagnosed (i.e., the darting date).
[d]Adult female rank or a cub's maternal rank the year during which the hyena was diagnosed. On the standardized rank scale, −1 corresponds with the lowest rank and 1 with the highest rank.
[e]Based on illegal livestock grazing in the park during the year in which a hyena was diagnosed. Here, we controlled for continuous age (mon) on the date of diagnosis because all cubs were from low livestock density areas.
[f]Significant at *P* value < 0.05.

population among cubs, as this was the subgroup with which we found an effect of *T. gondii* on approach distance to lions. First, we included the cubs' maternal rank as a covariate and noted no appreciable change in the estimate nor the interpretation of results (−2.14 [95% CI: −4.08, −0.21]). Next, we adjusted for the presence of a male lion during the hyena-lion interaction (−3.40 [95% CI: −6.03, −0.77]). Finally, we accounted for the presence of food during the interaction (−2.69 [95% CI: −4.91, −0.48]). Addition of the variables in our adjusted models did not markedly change the direction, magnitude, or precision of the estimate for *T. gondii* infection status in relation to approach distance.

Third, we explored associations of *T. gondii* infection with lion-related mortality. Among 33 mixed-age hyenas with known mortality causes, infected hyenas were nearly twice as likely to die by lions than by other known causes (52% vs. 25%). Infected hyenas were 3.91 (95% CI: 0.70, 32.78; *P* = 0.15) times more likely to die by lions than uninfected animals after accounting for sex, though this effect was not significant (Table 3). Among hyenas infected as cubs, 100% of the deaths were caused by lions, while only 17% of the deaths of hyenas not infected as cubs were caused by lions. In this small subsample of 11 hyenas infected as cubs, all of which were sampled between 1990-1999, the probability of dying by lions vs. other known sources of mortality was greater among infected than uninfected individuals (Fisher's Exact Test *P* = 0.01).

## Discussion

We found *T. gondii* infection was associated with behavioral boldness that brought infected hyena cubs into closer proximity to lions, as well as increased the likelihood of being killed by lions. The fact that we saw null and weaker associations among older (subadult and adult) hyenas suggests that experienced individuals might better assess threats and inhibit risky behavior, though other non-exclusive hypotheses could explain the observed age-dependent relationship between *T. gondii* infection and host behavior (Supplementary Table 2). Testing these models will require additional data and offer rich opportunities to advance our general insight to host-parasite interactions. Meanwhile, our results provide novel evidence of a widely reported but unproven

link between *T. gondii* infection, boldness toward felids under natural conditions, and fitness in a wild population of non-definitive hosts. This link is mechanistically plausible given that lions readily attack and kill hyenas[20,23] and are a leading source of hyena mortality in the wild[19]. Hyenas with above-average boldness in the presence of lions also have reduced longevity, which may result from injuries and lethal wounds inflicted by lions[21].

**Determinants of *T. gondii* prevalence.** Our analyses suggest that demographic factors influenced the prevalence of *T. gondii* in spotted hyenas inhabiting the Mara region. For example, older hyenas were more likely to be infected. A recent survey of carnivores from the Serengeti ecosystem found a similar pattern, suggesting that ingestion of infected prey or carrion—of which older individuals have a longer cumulative exposure—may be an important source of infection[24]. Counter to our predictions, *T. gondii*'s prevalence was not significantly higher within Masai Mara sampling localities characterized by relatively high livestock density. We initially predicted that hyenas, through ingestion of contaminated water or consumption of infected meat, would have a higher prevalence of *T. gondii* infection when living in close proximity to domesticated animals and human commensals since these are known parasite reservoirs[7,25]. However, we noted that *T. gondii* prevalence was stable across a gradient of hyena exposures to livestock densities (as well as multiple decades), indicating that this parasite has a widespread distribution throughout the Masai Mara region. Nonetheless, intensification of human activities may have important impacts on other variables (*e.g.*, the timing, genotypes, and outcomes of *T. gondii* infection).

**Possible mechanisms of behavioral alteration in *T. gondii*-infected hosts.** It is tempting to speculate that our findings reflect an underlying mechanism through which *T. gondii* manipulates hyena boldness to promote transmission to lions. However, several recent studies have also cautioned against over-interpreting observations from animals harboring both experimental and naturally occurring infections[26,27]. In light of important caveats raised by these other authors, our study was designed solely to

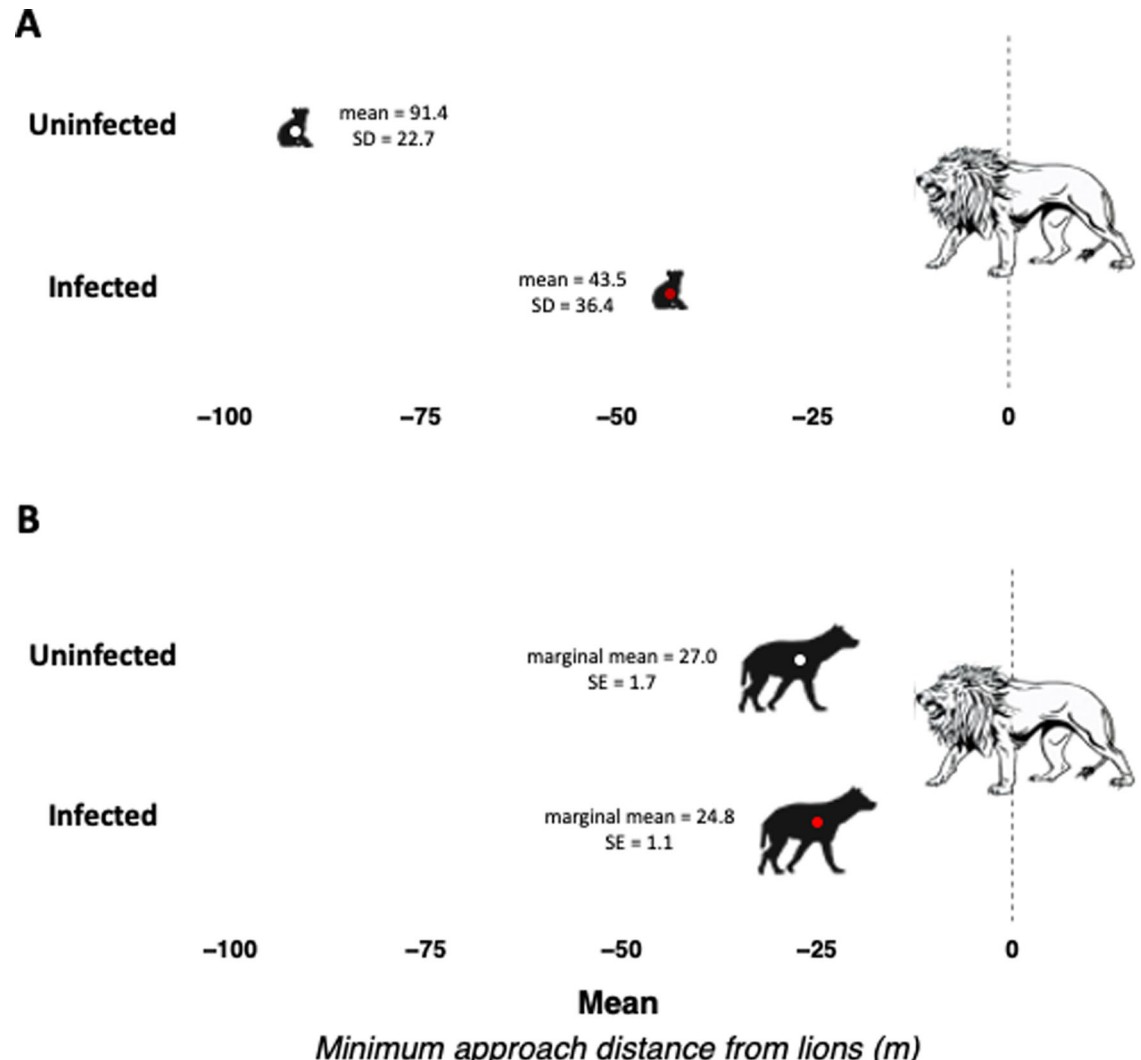

**Fig. 1 Mean minimum approach distance to lions for uninfected (white dots) vs. infected (red dots) hyenas. A** Among cubs ($N = 15$), estimates are raw means and standard deviations based on average minimum approach distance for the uninfected and infected. The cub data set includes only hyenas for which both diagnosis and distance from lions were measured as cubs. **B** Among subadults/adults ($N = 109$), estimates are marginal means and standard errors from a mixed-effects linear regression model that included *T. gondii* infection status as an explanatory variable, distance as a continuous outcome and a random intercept for hyena ID. Source data are provided as a Source Data file.

test one key prediction of the host manipulation hypothesis: that infected hosts will exhibit costly boldness in the presence of (definitive host) feline species. A second key prediction is that the alteration of host behavior is induced via adaptive parasite traits that have evolved to promote transmission; this prediction can neither be supported or falsified with our current dataset. Below, we discuss competing and plausible explanations for the observed association between *T. gondii* infection and fitness-related behavior in spotted hyenas.

In one scenario, *T. gondii* traits that facilitate transmission from a wide range of intermediate hosts (e.g., hyenas) into definitive hosts (e.g., lions) have evolved through natural selection on *T. gondii*. An ability to manipulate boldness in diverse hosts could be advantageous in biodiverse ecosystems like the Mara; propagation from definitive hosts (i.e., via oocysts) can infect a larger number of hosts over greater spatial and temporal scales in comparison to propagation via asexual stages (i.e., transmission between any two non-definitive hosts). However, an important caveat within the focal system is that lions rarely consume hyenas after killing them. Still, exchanges of blood and tissue during lethal conflicts may transmit *T. gondii* from an infected hyena to a lion where the parasite can sexually reproduce[28] (c.f. Supplementary Movie 1). Additionally, an infected hyena that is killed by a lion may also enhance parasite transmission because hyena carcasses are consumed by highly mobile carrion-feeding animals (e.g., vultures)[29] that can widely disperse the parasite (Supplementary Figure 4).

In a second scenario, the behavioral phenotypes of infected hyenas may simply represent "collateral manipulation" via traits that evolved to influence other host species like rodents. For instance, *T. gondii*-infected humans exhibit riskier behavior[7] despite being dead-end hosts. This possibility is further supported by findings that homologous neural and hormonal regulators of behavior are similarly altered by *T. gondii* infections in human and non-human hosts[30]. However, the concept of collateral manipulation has not been rigorously tested in wild animals, where researchers can directly assess its ecological and evolutionary significance. Only a small body of work examines *T. gondii*'s relationship to behavior outside of laboratory rodents and human hosts (e.g., sea otters and chimpanzees[5,13]); this work also examines a very narrow range of taxa despite the fact that a much wider array of mammals and birds are susceptible[7].

**Table 2 Associations of *T. gondii* infection with minimum approach distance to lion(s) among spotted hyena cubs (*N* = 15) and subadults/adults (*N* = 109).**

| *T. gondii* infection status | N | β (95% CI) minimum approach distance from lions | |
|---|---|---|---|
| | | **Unadjusted** | **Adjusted** |
| Cub hyenas (<12 mos)[a] | | | |
| Uninfected | 7 | 0.0 (Reference) | 0.0 (Reference) |
| Infected | 8 | −3.35 (−5.52, −1.18)[c] | −3.19 (−5.57, −0.81)[c] |
| Subadult/adult hyenas (≥12 mos)[b] | | | |
| Uninfected | 24 | 0.0 (Reference) | 0.00 (Reference) |
| Infected | 85 | 0.30 (−0.18, 0.78) | 0.27 (−0.19, 0.72) |

In all models, distances were square root transformed.
[a]Estimates are from a linear regression model where infection status is the explanatory variable of interest and minimum approach distance (m) from lions is the outcome. For one study animal, the minimum approach distance was an average of three repeated measures. Adjusted for sex and age (months) on the date of the hyena-lion interaction.
[b]Estimates are from a mixed linear regression model where the explanatory variable of interest is infection status, a random effect for individual ID, and the outcome is repeated measures of minimum approach distance (m) to lion(s). Adjusted for sex, age group on the date of infection diagnosis (subadult vs. adult), and age group on the date of hyena-lion interaction (subadult vs. adult).
[c]Significant at *P* value < 0.05.

**Table 3 Associations of *T. gondii* infection with odds of death by lion(s) among 33 hyenas.**

| *T. gondii* serostatus | N | OR (95% CI) of death by lion(s) vs. all other causes | |
|---|---|---|---|
| | | **Unadjusted**[a] | **Adjusted**[b] |
| Uninfected | 8 | 1.00 (Reference) | 1.00 (Reference) |
| Infected | 25 | 3.23 (0.62, 24.85) | 3.91 (0.70, 32.78) |

[a]Estimates are from a logistic regression model where the explanatory variable is infection status and the outcome is death by lion (yes vs. no).
[b]Model is adjusted for hyena sex (male vs. female).

A final scenario is that symptoms of *T. gondii* infection, such as encephalitis, coincidentally alter infected hyenas' behaviors but did not evolve through natural selection on *T. gondii* transmission. We cannot rule this out, though we note that if *T. gondii* caused severe neuropathy we would predict infections to elevate mortality involving other forms of behavioral maladaptation (e.g., lethal conflicts with Masai herders or vehicle collisions) and not just interactions with lions specifically.

This study is not without limitations. First, because we were only able to diagnose infection status at the timepoint when a hyena was darted and blood sampled, we were unable to assess how behaviors differed within individuals following *T. gondii* infection onset. Second, given the social nature of this species, the focal hyenas' behaviors towards lions during observation sessions were likely influenced by groupmates' behaviors, or the proximity of their mothers during these interactions. An ideal test of these predictions would involve hierarchical modeling that accounts for group composition and behavior during each observation session —an effort that is currently not possible with the available data. Finally, more sophisticated assessments of potential modes of transmission among livestock, pastoralists and hyenas are required to better understand how infection risks and outcomes are modified where humans and wildlife live in close proximity. These future directions will mandate heavier sampling and more sophisticated analyses that leverage longitudinal data to pinpoint the timing of *T. gondii* infections within hosts.

Our results suggest that behavioral boldness toward felines associated with *T. gondii* infection is likely deleterious to hyenas, at least when contracted early in life. We encourage further explorations of fitness-related behavior in natural settings—both for infections of *T. gondii*, which involve a substantial proportion of mammals and birds, and for other parasites suspected to alter host behavior to serve their own evolutionary interests.

## Methods

**The Mara Hyena Project.** This study uses data and samples from the Mara Hyena Project (approved by MSU IACUC and KWS), a long-term field study of individually known spotted hyenas that have been observed since May 1979. Study hyenas are monitored daily and behavioral, demographic, and ecological data are systematically collected and entered into a database. Here, we used data from four different hyena groups, called clans, as well as historic information about ecological conditions in the Masai Mara National Reserve. We maintained detailed records on the demographics of our study population, including sex, age, and the dates of key life-history milestones such as birth, weaning, dispersal and death. In the ensuing sections, we describe data collection and data processing procedures for assessment of *T. gondii* infection diagnosis, quantification of demographic and ecological determinants of infection status, and assessment of behavioral (boldness) and fitness (cause of mortality) characteristics hypothesized to be a consequence of positive *T. gondii* infection. The present analysis includes 168 hyenas, but specific subsamples vary depending on the particular hypothesis being tested.

**Biospecimen collection and assessment of *Toxoplasma gondii* exposure.** As part of our long-term data collection, we routinely darted study animals in order to collect biological samples and morphological measurements. Of special relevance to this study is our blood collection procedure. We immobilized hyenas using 6.5 mg/kg of tiletamine-zolazepam (Telazol ®) in a pressurized dart fired from a CO2 powered rifle. We then drew blood from the jugular vein into sodium heparin-coated vacuum tubes. After the hyena was secured in a safe place to recover from the anesthesia, we took the samples back to camp where a portion of the collected blood was spun in a centrifuge at $1000 \times g$ for 10 min to separate red and white blood cells from plasma. Plasma was aliquoted into multiple cryogenic vials. Immediately, the blood derivatives, including plasma, were flash frozen in liquid nitrogen where they remained until they were transported on dry ice to a −80 °C freezer in the U.S. All samples remained frozen until time of laboratory analysis for the *T. gondii* assays.

Using archived plasma, we diagnosed individual hyenas using the multi-species ID Screen® Toxoplasmosis Indirect kit (IDVET, Montpellier). This ELISA-based assay tests for serological (IgG) reactivity to *T. gondii*'s P-30 antigen and has been used in many prior studies of *T. gondii* in diverse mammals[22]. The output of the assay is an SP ratio, which is calculated as colorimetric signal of immunoreactivity for a tested blood sample (S) divided by that of a positive control (P), after subtracting the background signal for the ELISA plate (i.e., a negative control) from both S and P. We tested 168 plasma samples from 168 individual spotted hyenas and determined infection status based on the kit manufacturer's criteria for interpreting S/P: ≤ 40% = negative result, 40% < S/P < 50% = doubtful result, S/P ≥ 50% = positive result (Supplementary Fig. 1). Only 21 hyenas (13%) fell within the "doubtful" range of ELISA S/P values. We treated these individuals as negative in our analyses, following protocols from other recent studies[22]. Although ELISA-based assays performed relatively well in prior studies of *T. gondii* in both hyenas[31] and other mammals[22], the method can also be sensitive to cross-reactivity with antibodies to other, related parasites[32]. We therefore retested 60 randomly chosen plasma samples using an alternative, indirect fluorescence agglutination test (IFAT), which is less sensitive to cross reactivity[31], in order to confirm that the two methods yielded similar results using our plasma samples (Supplementary Fig. 2). Samples were submitted to the Michigan State University Veterinary Diagnostic Laboratory for a standard diagnostic IFAT procedure that used reagents supplied by VMRD, Pullman, WA. In brief, the IFAT measures the maximum dilution of a plasma sample at which immunoreactivity to *T. gondii* antigen is visible by microscopy. Two samples were excluded from our main analyses because of suspected assay error (e.g., one negative SP ratio and one additional IFAT vs. SP ratio discrepancy) making our final diagnostic sample size, *N* = 166 hyenas. However, it should be noted that inclusion of these two questionable data points did not substantively change our results and had no effect on our analyses of hyena boldness or fitness as these two hyenas lacked data required for those analyses. To rule out the possibility of misclassification of *T. gondii* infection due to titer decay over time, we plotted the SP ratio for infected and uninfected animals separately with respect to age and did not observe any decline in SP ratio as a function of age (Supplementary Fig. 3). Diagnostic assays were performed by people who were blind to the individual hyena's demographic, ecological, and behavioral data.

**Demographic, social, and ecological characteristics.** The first aim of this analysis was to identify demographic and ecological correlates and determinants of *T. gondii* infection. The key characteristics of interest include sex and age, two

demographic traits that have previously been implicated in health and behavioral outcomes, as well as exposure to livestock density.

We determined the sex of each hyena based on the glans morphology of its erect phallus during field observations; this is reliable starting at 3 months of age[33].

We aged each hyena by back-calculating its birthdate based on its physical appearance when first observed in infancy. Based on its pelage, morphology and behavior, we are able to determine a birthdate with an accuracy of ±7 days[34]. We used this method to determine each hyena's age in months at the time of blood collection. In the analysis, we assessed age continuously in months, as well as in distinct age groups divided into cubs (<12 months), subadults (≥12 to ≤24 months), and adults (>24 months of age). The age cut-offs were determined based on the timing of major life history milestones; weaning occurs at approximately 12 months of age and hyenas of both sexes achieve reproductive competence at around 24 months of age[34,35].

As part of our routine data collection, all aggressive interactions between hyenas are recorded, such that we can calculate rates of threat displays, chases, and bites between clan members. Agonistic interaction data are used to calculate each hyena's dominance rank each year via a matrix of dyadic wins and losses in fights[36–38]. Values from each rank matrix are normalized as a continuous variable from −1 (lowest) to 1 (highest) and are updated annually to account for demographic change. We use maternal rank as a proxy of each cubs rank until they learn their own rank.

To quantify exposure to livestock density, we took advantage of naturally occurring variation in exposure to human activity across distinct regions of the Masai Mara National Reserve. We classified all hyenas from the western region of the Reserve (also known as the Mara Triangle) as "low livestock density" due to strict bans on livestock grazing and travel on foot in this area. The eastern side of the Reserve borders pastoralist villages that have experienced an increase in human population growth in recent years, especially around the burgeoning Talek community[39]. Additionally, assessment of trends in livestock counts within the Reserve indicates a marked increase in illegal livestock grazing on the eastern side of the Reserve starting in 2000, followed by another increase between 2009 and 2013. These changes coincided with parallel shifts in hyena demography and wildlife community composition[40]. To improve discriminatory power in our analyses that reflect these changes in livestock density and shifts in ecology, we enriched our sample selection to include hyenas from areas of low and higher livestock density. For "low livestock density," we selected animals from the eastern side of the Reserve sampled before 2000 along with animals from the western side of the Reserve (any time period). For "high livestock density," we selected animals from the eastern side of the Reserve from 2012 onward.

**Boldness behaviors and fitness.** In addition to identifying determinants and correlates of *T. gondii* infection, we also sought to explore the effects of infection status on hyena behavior and fitness. Over the duration of our study, we documented all observed hyena-lion encounters i.e., all instances where at least one hyena and at least one lion approached to within ~200 m of one another. In 731 observation sessions we recorded 3791 minimum distance estimates between individual hyenas and one or more lions along with the date, location, and identities of all hyenas present, as well as whether food (a dead prey animal or its components) or a male lion was present during the encounter, as both these factors are known to influence hyena behaviors. All boldness behaviors were extracted by four individuals blinded to infection status with 83% agreement across seven metrics recorded during hyena-lion interactions[41].

Based on previous findings in this study population that minimum distance from lions is a measure of boldness that shows inter-individual consistency and corresponds with fitness[21], we used this metric as an indicator for behavioral boldness. During each hyena-lion encounter, we recorded the distances between lions and individual hyenas in meters using 20-min scan sampling of individual hyena distances from the nearest lion, as well as all-occurrence sampling of close behavioral interactions between lions and hyenas (e.g., a hyena comes within 10 m of a lion). Because the body length of an adult hyena is ~1 m, we are able to accurately estimate approach distances at this scale. Due to the inherent frenetic activity at some hyena-lion encounters, some of the minimum approach distances were recorded as ranges (e.g., 10–15 m) or inequalities (e.g., <10 m). For ranges, we calculated the mid-point and used this value in the analysis (e.g., if the range was 10–15 m, then we used 12.5 m in our calculations). If the distance range was large (>25 m) and therefore highly uncertain (i.e., if the range exceeded ½ a standard deviation as estimated from the minimum approach distance data set), we removed it from the dataset. Of the 529 approach distance ranges in our data set, 225 were removed because the range exceeded 25 m. We retained inequality distances by using the 'less than' distance if the recorded distance was smaller than 25 m (approximately the mean [mean = 45 m] minus ½ a standard deviation [sd = 50 m] of all hyena minimum approach distances) and by including the 'greater than' distances if the recorded distance was greater than 75 m (approximately the mean plus ½ a standard deviation of hyena minimum approach distances). For example, a distance recorded as <50 m would be removed from the data set as it could include a wide range of actual distances (0–50 m), while a recorded distance of <15 m was retained in the data set as 15 m. As a result of filtering inequality distances with large uncertainty, we removed 67 of 72 approach distances recorded as inequalities. Finally, we filtered the hyena approach distance to lions by

removing instances when the minimum approach distance exceeded 100 m, given that at this range hyenas and lions pose little threat to one another. After filtering, our final data set included 2725 minimum approach distance estimates. It should be noted that during any particular hyena-lion interaction, we retained a single minimum approach distance for each hyena, but over their lifetime hyenas interact with lions on multiple occasions, thus the repeated minimum distance measures for individual hyenas.

Our longstanding behavioral database also documents the source of mortality for each hyena whenever known. Deaths attributed to lions included cases in which lions were observed killing hyenas and when fresh corpses of hyenas were found with puncture wounds in each corpse made by canine teeth that were too far apart to have been inflicted by anything but a lion. In our analysis, we dichotomized cause of mortality as death by lion vs. all other known causes of mortality and evaluated this as a binary outcome in the statistical analysis. We did not include data in which a hyena's cause of death was unknown.

**Statistical analyses.** In the analysis, we tested three hypotheses: (*H1*) higher livestock density is associated with higher risk of *T. gondii* infection in spotted hyenas; (*H2*) infected hyenas behave more boldly towards lions than uninfected hyenas, as indicated by a shorter minimum approach distance; (*H3*) *T. gondii* infection imposes fitness costs on the host, as indicated by greater odds of death by lion(s). We describe methods for testing each hypothesis below, following a description of our general approach to data analysis.

Prior to formal analyses, we assessed the distributions of all variables. This included viewing the distributions and calculating descriptive statistics for continuous variables (e.g., minimum approach distance towards lions) to check for deviations from normality and missing values. We also assessed frequency distributions for all categorical variables (infection status, sex, age group, food presence, and livestock density). Finally, as part of our data exploration, we conducted bivariate analyses of associations between demographic and ecological characteristics and infection status, as well as associations of demographic or ecological characteristics with behavioral outcomes to identify covariates for inclusion in multiple variable analysis. In all models, we considered an estimate to be statistically significant at a nominal cut-off of alpha = 0.05. Data cleaning and analyses were performed in program R version 4.0.2[42]. Linear mixed models were conducted using the lme4 package[43], version 1.1.21.

H1: Greater livestock density is associated with higher risk of *T. gondii* infection. In this portion of the analysis, we used univariable logistic regression to investigate the relationship between livestock density (high vs. low) as the primary explanatory variable of interest, and *T. gondii* infection (positive vs. negative) as the outcome. In addition, we also explored associations of other key demographic characteristics as determinants of infection, namely sex, age at diagnosis, and social dominance rank. Following the simple regression models that contained one single explanatory variable (unadjusted analysis), we also examined multiple-variable (mutually-adjusted) associations among the above variables. In models where dominance rank was not the primary variable of interest, we did not include rank as covariate due to missing data.

H2: Infected hyenas behave more boldly towards lions, as indicated by shorter minimum approach distances. To investigate the extent to which infection status is related to boldness behaviors, we used simple (unadjusted) and multiple-variable (adjusted) linear regression models in which *T. gondii* diagnosis (infected vs. uninfected) was the explanatory variable of interest, and the hyenas' square root transformed minimum approach distance (m) was the outcome. We transformed the distances to improve assumptions of normality. We stratified all models by age group such that cubs were analyzed separately from subadults and adults. We made this decision based on bivariate associations that revealed a significant age structuring of infection status (i.e., much higher prevalence of infection in subadults and adults than in cubs), as well as significant effects of age on hyena approach distances towards lions (i.e., older hyenas were much more likely to approach lions closer than younger hyenas). The cub models included individual hyenas that had both infection diagnosis and hyena-lion interaction data during their first year of life. Similarly, the subadult and adult models were restricted to include only infection diagnosis and hyena-lion interactions collected from hyenas 12 months of age and older.

When exploring associations among cubs, we first examined the unadjusted association of *T. gondii* diagnosis with minimum approach distance to lions. Next, we controlled for sex and age in months on the date of the interaction with lions. The age distributions of hyena cubs during observed interactions with lions were 2.7–8.5 months for uninfected cubs and 3.2–11.8 months for infected cubs. We did not need to account for livestock density because all cubs were sampled in low livestock density areas. Additionally, for all but one cub, we only had a single minimum approach distance from lions. For the cub with multiple measures (N = 3), we took the average of its minimum approach distances for use in the analysis.

When exploring associations among subadults and adults, we used a similar modeling strategy to that used for cubs, except rather than using conventional linear regression, we employed mixed linear regression models to account for the multiple assessments of minimum approach distance to lions for hyenas in these two age groups (median = 5 measurements per hyena) via a random intercept for the hyena's ID. After examining unadjusted associations, we implemented a multiple variable model that adjusted for age group (subadult vs. adult) both at the

time of the diagnosis and at the time of the hyena lion interaction, and sex (male vs. female). *Nota bene*: in the subadult and adult model age was not parametrized as a continuous measure (e.g., age in months) because for some adult female hyenas, who we began observing as adults, and for some immigrant males, whose natal clan is not known, we do not know the exact birth date of these hyenas.

In addition to the above analysis for subadult and adult hyenas, which leveraged all available hyena-lion interaction data, we also conducted sensitivity analyses on a restricted data set wherein we only considered approach distances from lions that occurred prior to the diagnostic date among hyenas who tested negative for *T. gondii* infection, thus ensuring these represented behaviors of uninfected hyenas. Similarly, we only considered hyena-lion interaction data that occurred after the diagnostic date for individuals who tested positive for *T. gondii* infection. The rationale for this approach is rooted in achieving temporal separation to avoid erroneously examining hyena-lion interactions for negative diagnosis hyenas who subsequently became infected and vice versa (*nota bene*: we did not do this for cubs given our small sample size in this age group and because the small age range limited the possibility that a hyena's approach from lions did not reflect its infection diagnosis). Using this restricted dataset, we modeled the associations between infection status and each hyena's closest approach distance to lions following the previously described modeling strategy. We also modeled the hyena approach distance from lions as function of *T. gondii* infection among hyenas diagnosed as either positive or negative but excluding the doubtful diagnosis category. This second sensitivity analysis aimed to rule out any potential variable misclassification bias.

H3: *T. gondii* infection imposes fitness costs on the host, as indicated by greater odds of death by lion(s). Here, we assessed the probability that *T. gondii* infection in hyenas was associated with lion-induced mortality. To do this, we used logistic regression models to compare the odds of mortality due to lions vs. all other known causes of mortality for infected vs. uninfected hyenas. Following unadjusted analysis, we controlled for sex in a multiple-variable logistic regression analysis. Due to small sample sizes (i.e., cells in cross tabulations with $N = 0$) we were not able to adjust for hyenas' ages and livestock density levels. However, we were able to use a two-by-two table and Fisher's exact test to determine whether the probability of dying by lions vs. other sources of mortality differed between infected and uninfected cubs.

Additional sensitivity analyses. Dominance rank, presence of food, and presence of male lions are key determinants of boldness behaviors in hyenas. Therefore, in age-stratified subgroup analyses where *T. gondii* infection was a significant determinant of approach distance to lions (i.e., in cubs only), we included maternal rank, presence of food, and presence of male lions to rule out the possibility of extraneous causes of boldness behavior. We then assessed the extent to which inclusion of each of these variables, singly, changed the direction, magnitude, and precision of the estimate for *T. gondii* infection in relation to approach distance to lions. Similar to cubs, we ran sensitivity analyses that included food present during the interaction (yes vs. no), and livestock density during the year of the interaction (high vs. low).

**Reporting summary**. Further information on research design is available in the Nature Research Reporting Summary linked to this article.

## Data availability
The full data used in this paper are available at https://zenodo.org/badge/latestdoi/245864615 (https://doi.org/10.5281/zenod.4699720)[44]. Source data are provided with this paper.

## Code availability
Source code is provided with this paper at https://zenodo.org/badge/latestdoi/245864615 (https://doi.org/10.5281/zenod.4699720)[44].

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

## Acknowledgements

We thank the Kenyan National Commission for Science, Technology, and Innovation, the Naboisho Conservancy, the Narok County Government, the Kenya Wildlife Service (KWS) and Brian Heath for permission to conduct research in the Mara ecosystem. We also thank Chiara Bowen and Nichole Grosjean for lab assistance, Samantha Gregg, Katie Keyser, Leah McTigue, and Abigail Thiemkey for assistance extracting distances between lions and hyenas, the NSF-BEACON center for the Study of Evolution in Action for funding, the Mara Hyena Project field crew, and the residents of the Mara ecosystem for their direct and indirect support of long-term field research in their backyard. We thank Joy Baldwin for providing the lion illustration and Mark Bellncula and Page E. Van Meter for providing the hyena silhouettes seen in Fig. 1. We thank members of the Getty and Holekamp labs for their feedback on this project and paper. This material is based in part upon work supported by the National Science Foundation under Cooperative Agreement No. DBI-0939454. Any opinions, findings, and conclusions or recommendations expressed in this material are those of the author(s) and do not necessarily reflect the views of the National Science Foundation. Sample collection and behavioral data collection were supported by NSF grants IOS1755089 and OISE1853934. Z.M.L. was funded by the Morris Animal Foundation grant D19ZO-411.

## Author contributions

Conceptualization, E.G., Z.M.L.; data generation and curation, E.G., Z.M.L., P.W., G.S.H., K.D.S.L., T.M.M., J.W.T., M.O.P.; formal analysis, Z.M.L., W.P.; funding acquisition, E.G., Z.M.L., K.E.H.; investigation, E.G., Z.M.L.; methodology, E.G., Z.M.L., P.W., G.H., K.D.S.L., T.M.M., W.P.; resources, E.G., P.W., G.S.H., K.E.H.; supervision, K.E.H., T.G.; visualization, E.G., Z.M.L.; writing—original draft, E.G., Z.M.L.; writing—review & editing, E.G., Z.M.L., P.W., G.S.H., K.D.S.L., T.M.M., J.W.T., W.P., M.O.P., K.E.H., T.G.

## Competing interests

Authors declare no competing interests.
