## [Peer Review File · Nature Communications]

REVIEWER COMMENTS

Reviewer #1 (Remarks to the Author):

Review of the article entitled: *Toxoplasma gondii* infections are associated with costly boldness toward felids in a wild host

Overall, I found the results presented very interesting (parasite manipulation is a fascinating topic not well explored in natural systems), although the manuscript deals with two main findings and thus appears a bit limited. In addition, one main finding lies on a very small sample size (N=15 cubs) and we cannot totally exclude spurious associations between minimum approach distance and infection status in cubs. However, this result based on cubs' behavior is reinforced by a second finding about the probability of dying from lions as a function of individual infection status (which I found very nice!). I leave thus the editor deciding whether the results presented deserve publication in Nat Comm.

I only few more or less minor comments (organized by line number).

-L87: did you re-run your analyses excluding those samples with an uncertain diagnose? I think it would be worth.

-L90: the "r" coefficient is quite low (% of variance explained: 49%), why is that? Also, did you exclude the obvious outlier from your sample set (Figure S2)?

-L95: I found somehow unexpected that dominance rank did not influence *T. gondii* prevalence because I would have (perhaps naively) expected a relationship between boldness and dominance. Is it not the case in your study system that more dominant individuals are also more bold?

-L100: I'm wondering whether authors should present only the results from adjusted analyses. Unadjusted analyses do not bring extra information.

-L120: to increase sample size in cubs, I'd suggest performing analyses using sliding age-windows. For example, authors could analyze individuals aged 18months or less; 24 months or less.

-L386: is it possible to quantify the intensity of infection? It would be interested to correlate this quantitative measurement with boldness.

-L495: 'all other causes of death' did not include those 'ambiguous' deaths that could have been attributed to lions, right?

-L554: I don't understand why, in the subadult-adult model, authors did not consider age as a continuous variable. I think it is important to control for age in months as done for cubs, especially because infection status depends on age.

Reviewer #2 (Remarks to the Author):

Gering et al., take a novel approach, by using data from a wildlife system, to test the general ecological hypothesis that certain pathogens have evolved to cause 'risky' behavior in their host species to

promote transmission. Their findings suggest that hyenas infected with *T. gondii* show more risky or bold behavior, and that this behavior may drive transmission of the pathogen from intermediate to definitive host. For example, they show that infected hyena cubs approach lions more closely and are more likely to die by being killed by a lion. This is a fascinating topic and it's exciting to see these hypotheses tested in a wildlife system. The authors do an excellent job of formulating a number of interesting hypotheses and providing a detailed discussion of possible mechanisms underlying their findings. However, I had some concerns regarding both the statistical and diagnostic approaches (e.g. determination of an infected/uninfected status in individual hyenas) that need to be resolved in order to have confidence in the authors reported results/findings.

Comments and suggestions:

- Regarding infection status, older animals may be infected, but titers may have declined below detection, so even if they test seronegative, they may have been exposed in the past.
- The authors could assess whether the young animals always had relatively high titers, and if so, one could potentially assume that titer decline is slow enough that if an animal is < a certain age, then the failure to detect a titer means that the animal was never exposed (vs. exposed a long time ago and the titer has declined below detection). Alternatively, the authors could look to see whether there are sufficient data from known positive animals through time that a titer decay rate may be estimated. This could then be used to estimate the time that it takes a titer to decline below detection after initial exposure and only animals younger than that could be included in the analyses to increase the probability that seronegative animals are truly uninfected.
- Statistical approaches – rather than performing multiple 'univariable' or bivariate analyses, I would strongly recommend including all potential explanatory variables in the model and using some sort of model selection technique to arrive at the best fit model. I was confused as to why age was used as a categorical variable in some analyses and as a continuous variable in others. I also think (but certainly the authors should consult with a biostatistician on this – as with other statistical issues) that there is no need to perform separate analyses on cubs vs. the older age classes. I believe that this would be captured by a single mixed effects model that includes age*infection status interactions and individual ID as the random effect.
- Statistical language – throughout the MS the language describing the statistical approaches struck me as unconventional. I would strongly recommend including a biostatistician as a coauthor so that they can help write the statistical sections and advise on the statistical approaches. E.g. the authors refer to a sensitivity analysis, but this seemed to be an analysis of the contribution of other possible explanatory variables.
- Livestock density and human disturbance are used interchangeably, I would suggest choosing one, defining it and then staying consistent throughout the MS as to how this variable is referenced.
- H1 this is an interesting hypothesis, but I'm not entirely convinced of the mechanism. I think the biological mechanism underlying this should be more clearly explained. As it stands, it doesn't make a ton of sense to me unless the hyena are preying on livestock and there's a high prevalence of *T. gondii* infection in livestock, or if domestic cats are accompanying these illegal livestock grazing events (which I don't think they are...).

- The authors repeatedly mention adjusted vs. unadjusted and I found this very confusing. What these two terms refer to should be stated more clearly. I believe they mean models including just the explanatory variable of primary interest vs. those which include all possible explanatory variables.
- The authors mention using a restricted vs. unrestricted data set for the analyses based on the plasma *T. gondii* ELISA test results. This is fantastic! I would suggest that only the data set that they refer to as 'restricted' should be used. Or at least limit the length of a time interval between negative test results and behavioral observations. Given the high seroprevalence of anti-*T. gondii* antibodies in the hyena population and the fact that probability of infection increases with age, it's very possible that new infections could occur if the time interval between plasma collection and behavioral observation is long. If the authors wanted to consider including behavioral observations from a time period prior to a positive test result, one approach would be to consider the actual titer magnitude, and if available, calculate titer decline in their hyena population using data from animals for which multiple positive test results are available. If animals have a very high titer, this is strongly suggestive of a recent infection. If animals have a low titer and both the rate of titer decline and the expected maximum titer for that population can be estimated, then it may be possible to back calculate the time of exposure (although this is a very tricky process and there may not be sufficient data to attempt this).
- For the 'death by lion' analyses – I think the authors could use a similar glmm approach as outlined above but using a logistic regression instead of a linear regression with death by lion vs. death from other cause as the outcome variable. In this case as well, I think it would be important to take into consideration the time between death and the negative test result as these animals could have become infected in the interim.
- Some people will analyze data using multiple cutoff values and run two sets of analyses, one which includes the 'indeterminant result' animals and one which excludes them from the positive 'bin'. The authors may want to consider running such analyses to assess whether altering the cutoff to include the lower SP ratio animals changes their findings.

Fig. S1. This is a great figure and presents the data clearly; however I found the descriptions in the text part to be a bit confusing. You may want to consider stating "Distribution of *T. gondii* ELISA results (SP ratios) from spotted hyena plasma samples collected from animals in or near". It may also be a bit clearer to elaborate by saying something like "The dashed red line is the upper SP ratio cutoff for negative diagnoses, i.e. ratios < XX are considered negative, and the solid red line is the lower cutoff for positive diagnosis, i.e. all ratios > 0.5 were considered positive." You could also simplify by just say something like "SP ratios below the dashed red line were considered negative, those above the solid line were considered positive and those between the lines were indeterminant".

Fig. S2. Please include what is considered a positive result for the IFAT. This is important as the IFAT results seem to indicate that there was detectable fluorescence at some dilution in all samples. If the cutoff between positive and negative is fluorescence at any dilution vs. no fluorescence, then it seems that all of these samples are positive. This would obviously make any subsequent analyses impossible. Also, I'm not sure if this is the right analysis to perform in order to assess comparability between tests. A Cohen's Kappa (McHugh ML. Interrater reliability: the kappa statistic. Biochem Med (Zagreb).

2012;22(3):276-282) may be more appropriate with test results binned into positive and negative categories, but I would consult with a biostatistician to confirm this. If the correlation is the correct test to use, I'm still not sure that a correlation of 0.7 indicates good agreement, but again, best to consult with a statistician about this. There are a few other diagnostic test issues to consider: for the ELISA, is the positive control a hyena positive control? I believe that the IFAT is generally considered to be the diagnostic test of choice for determining serum (or plasma) antibody titers, so it would be good to be able to provide strong evidence that the ELISA results are roughly equivalent. But I also found this article where the authors found good agreement between an ELISA and IFAT for detecting anti-T. gondii antibodies (Glor, S.B., Edelhofer, R., Grimm, F. et al. Evaluation of a commercial ELISA kit for detection of antibodies against Toxoplasma gondii in serum, plasma and meat juice from experimentally and naturally infected sheep. Parasites Vectors 6, 85 (2013). <https://doi.org/10.1186/1756-3305-6-85>). They also used the Kappa to assess agreement between tests, so perhaps a good reference.

Table S1. I would suggest reporting the SD vs the SE so that the reader can more readily assess the variation in the data.

I found the superscript descriptions confusing, I would suggest working to clarify these. For example, instead of "From an independent t-test for sex, food presence, and human disturbance; from a Wald chi-squared test for age group" perhaps have one superscript indicating that the analysis/p-value was from an independent t-test, and then mark the appropriate corresponding values, similarly for the Wald chi-square.

Food presence and whether the lion was an adult male were not included in this table, I would suggest including these even if they are not significant predictors of distance. Otherwise, it's confusing that these explanatory variables are mentioned in the superscripts but are not present in the table.

There were typos and grammatical errors in the C superscript: I think it should be "cubs' maternal..." also interaction, not interact.

Also, a mean of -5.14 for Dominance Rank doesn't make sense. Do the authors mean a reduction in the mean minimum approach distance of 5.14 with each 1 unit increment in standardized rank? If so, please adjust accordingly. Or perhaps I'm not understanding what's being reported in this table?

Movie S1. This is a really interesting video! I'm wondering though whether this sort of interaction would really represent a significant transmission risk. Infection through blood seems a very unlikely scenario. Toxo encysts in muscle and neural tissue and these tissues would be the most likely source of infection during a lethal lion-hyena interaction. However, if lions aren't eating the hyenas either, it seems that transmission through consumption of muscle or neural tissue is also unlikely. But I'm not a T. gondii expert, so perhaps best to consult with one. If this is a likely transmission scenario, then I would recommend including a reference as to support the fact that transmission can occur through these routes and/or through ingestion of even a very small amount of infectious material.

Results

I'm concerned that given the very high SP of anti-T. gondii antibodies that the adults that are negative

are not true negatives, but rather animals with titers that have dropped below detection. An assessment of antibody titer changes through time in single individuals could help provide insights into whether this is an issue or not. I.e. if titers remain at fairly constant levels post exposure, then this wouldn't be an issue, but if *T. gondii* titers decline through time, since we know the protozoa isn't cleared, this would suggest that some of the apparently negative animals (especially those falling within the 'indeterminant SP ratio range) may actually just be animals that are infected with *T. gondii*, but that infection occurred long enough ago that antibodies are no longer detectable. One thing that could be done with the current data would be to assess whether the SP ratio correlates with age. Presumably, animals with more recent infections will have higher titers. So, if infection risk increases with age, then younger positive animals are likely to have been exposed more recently than older positive animals. Therefore these younger animals may have higher titers, while older animals that may have been infected many months or years prior, may have lower titers. If the authors find this to be the case, then they should consider the possibility that the test negative older animals may in fact be infected but seronegative. It is also worth doing a literature review to assess whether there is evidence of this phenomena occurring (i.e. seronegative animals that are confirmed positive after necropsy and PCR or special staining of tissues).

Table 1.

Typos in superscript d: I think it should be cub's maternal (not cubs matnema)

Figure 1. The mean +/- SD would be more informative, I would recommend changing accordingly. I also don't understand why the hyena age group on the date of diagnosis would be relevant, assuming that this date occurred prior to the date of the hyena-lion interaction. I would think only the age of the animal at the time of interaction would be relevant.

Table 2. Tables should be able to stand alone without looking at the text. I would suggest clarifying what is meant by adjusted and unadjusted, as written in the table, it's very confusing. In the MS text (Lines 123-126) it is more clear.

Lines 127-135: I believe that a sensitivity analysis is when you look at how variation in a given variable impacts output, not how addition of new variables impacts output. I think what the authors are doing here is assessing the significance of various additional potential predictor/independent variables on the dependent or outcome variable (i.e. minimum distance to lion). If this is the case, then I think a more appropriate approach would be to determine the best fit model by including all potential predictor variables in the glm and then use a model selection process to determine the best fit model. I was also a bit confused by the wording of the last sentence. I think the authors mean that the model estimated the approach distance as a function of *T. gondii* infection status (as determined by serostatus), age, sex etc., this is not clear as currently worded.

Lines 150-151: I think a chi-square or Fisher's exact should be used to assess the statistical significance of this difference between infected and not infected (i.e. 52% vs. 25% died from lions vs. other causes). Or, as I suggested above, a logistic regression could be used for these analyses.

Lines 153-158: I found this a bit confusing. I wasn't sure what the relationship between the statement in lines 155-156 was to the last sentence of this paragraph. Please clarify.

Table 3. Same question as mentioned above regarding the use of the term 'adjusted'. Please clarify what is meant by 'model is adjusted for sex'.

Lines 168-171: an alternative hypothesis may be that essentially all subadult and adult hyenas have been infected, but some no longer have detectable titers.

Methods

Minimum approach distance to lions: The authors included data if hyena's distance from lion is >75 m and they include it as 75 m. It seems that this could significantly alter the calculated mean depending on the true value. Perhaps it would be best to bin all animals $100 > X > 75$ (regardless of whether the exact number is known or not)?

Death by lion: How did the authors deal with deaths due to unknown causes, i.e. could have been a lion but the carcass was too damaged to determine?

STATISTICAL ANALYSES

Rather than performing individual linear regressions for each potential explanatory variable, I would suggest including all in the model and performing some sort of stepwise process to remove those that don't contribute significantly to the model fit. There are a number of ways to do this, I would recommend consulting with a biostatistician to determine the best approach for your data.

In discussion section the authors mention additional future analyses – I would recommend that they consider some of these for this MS. Especially the ones that consider the number of other hyenas involved in the hyena – lion interaction as well as their behavior. I imagine hyena boldness would increase as the number of additional hyenas increases. Similarly, I think it is important to note the number of lions involved in the interaction. Again, I imagine as the number of lions involved in the interaction increases, the boldness of the hyenas would decrease. I don't know what the most reasonable statistical approach would be to assess this, likely some sort of hierarchical Bayesian framework (as the authors mentioned), but given all of the factors that could have a biologically important and statistically significant impact on hyena boldness, I think it's important to consider and include all of these (or select interactions that control for this, e.g. only include single hyena, single lion interactions in the analyses).

Reviewer #3 (Remarks to the Author):

This well-written manuscript describes an innovative and longitudinal study showing higher boldness in *Toxoplasma gondii* infected hyenas (intermediate carnivore/carrion host) toward one of the definitive host of the parasite in the wild; the African lion. Beside a couple of clarifications needed in the Methods and a couple of points to consider in the interpretation of the results, my main concern lies in the 'novelty' of the manuscript given the authors previous publication: "Times makes you older, parasites make you bolder – *Toxoplasma gondii* infections predict hyena boldness toward definitive lion hosts" in Banzhaf, W., Cheng, B. H. C., Deb, K., Holekamp, K. E., Lenski, R. E., Ofria, C., ... Whittaker, D. J. (Eds.). (2020). *Evolution in Action: Past, Present and Future*. Genetic and Evolutionary Computation.

I understand that the nature and motives of both writings may differ but the study is essentially the same. As stated in the book chapter, the authors seem to consider the submitted manuscript as a follow-up study. If so though, it is strange to me that they do not even refer to their pilot, particularly for the serology results. In the submitted manuscript, the authors tested three hypotheses: (H1) higher livestock density is associated with higher risk of *T. gondii* infection in spotted hyenas; (H2) infected hyenas behave more boldly towards lions than uninfected hyenas, as indicated by a shorter minimum approach distance; (H3) *T. gondii* infection imposes fitness costs on the host, as indicated by greater odds of death by lion(s). In the previously published book chapter, H1 and H2 were also tested and provided similar results (for H1) and more refined ones (for H2; i.e. different models for cubs and sub/adults).

One difference may reside in the narratives. While in the former writing, the authors introduced the concept of 'collateral manipulation' – as they judged the parasite transmission unlikely to occur between hyenas and lions due to the lack of consumption by the latter – they revised their theory in the current manuscript adding the fact that blood and tissue ingested by lions may also transmit the parasite.

In sum, the authors should clarify what new input this manuscript brings compared to their recently published book chapter.

L54-55 Introduction. It may be necessary to remind readers how *Toxoplasma gondii* can be transmitted.

The Introduction would be the place to introduce the authors' pilot study, and the novelty of the current study.

L84 Serology results for IgG to *T. gondii*: These seem essentially the same data and findings as in the authors' previously published book chapter, regarding the number of tested individuals (168 – 2 discarded samples = 166 vs. 168 previously); the reported number of seropositive (109 vs. 110) and seronegative (36 vs. 37) individuals; as well as doubtful diagnostics (21 in both writings). The only new addition I see here is the use of IFAT diagnostics for a subset of the samples to compare with ELISA,

showing high reliability. Since a larger dataset does not seem to have been used for these analyses (compared to Gering, Laubach et al. 2020), the authors should make clear what differs – if anything – in terms of results, and if not, simply refer to/cite their previous publication for this part of the results.

L90 If the data (IFAT & ELISA) do not follow normality, it may be better to use Spearman rank correlation or Kendall rank correlation instead of Pearson correlation. Spearman and Kendall would rank all the values and correlate the ranks. They may also better account for potential outliers in this subset of samples that can drive correlation coefficients using Pearson.

Results. I would recommend adding subheadings that refer to the three main hypotheses (see Statistical analyses in the Methods) being tested.

L129-130 Did the authors test whether there was a correlation between the infection status of the cubs and the one of mothers? Given the transmission of the parasite, one could predict that times of exposure to the parasite could be similar between mothers and cubs given their proximity. In addition, how far were mothers from the cubs when a lion was in sight? We may also expect a correlation between the minimum approach distance exhibited by cubs and mothers, i.e. mothers of infected cubs getting closer to lions. This might be something to add in the discussion – whether infected cubs were on their own when exhibiting closer distances to lions or were there other individuals around. In other words, I was wondering how was the social dynamic like during these approaches. On the same line of thought, I was also wondering if infected mothers would be more permissive/less protective toward their cubs compared to uninfected mothers.

L146 Between Table 1 & Figure 1 / Table 2, it may be helpful to remind why there is a drop of $49 - 15 = 34$ cubs in the analyses. It should be added in one of the table's/figure's legend that 15 is the number of (infected/uninfected) cubs for which proximity data with lions were available.

Table 3. The difference in probability of dying from lions vs. other causes between infected and uninfected individuals could also be due to the inequality of known causes of death between these two groups (N=25 for infected individuals and N=8 for uninfected ones).

Discussion. It may be interesting to reflect on how lions may want to avoid the parasite – if costly to them – by e.g. avoiding consuming infected individuals / having evolved recognition mechanisms of potential sources of infection.

LL173-178 Have lions been tested positive to *T. gondii* in the Masai Mara? One of the next logical step, as suggested above, would be to investigate whether the susceptibility of infection in lions is correlated with their killing/predatory behavior.

LL227-228 The study by Poirotte et al. on *T. gondii* infected chimpanzee's attraction toward predator urine was performed with captive chimpanzees and thus the term 'wild hosts' here is confusing and inappropriate.

Perspectives: How infected hyenas might be generally less responsive toward (i.e. show higher latencies to approach) a simulated threat (see Turner et al. on field experiments with a simulated hyena) is something important to reflect on and lacking from this manuscript. This previous result (Gering, Laubach et al. 2020) converge with the hypothesis that infected hyenas may be less risk sensitive, disregarding the origin of the treat.

L388 Methods. It would be useful to state here that one plasma blood sample was tested per hyena.

LL445-448 How many hyenas were sampled for each low and high livestock density areas? The same samples as in Gering, Laubach et al. 2020 seem to have been used although partitioning seems different – if so, the authors should mention the number of samples collected for each locality and/or refer to their previous book chapter.

L537 Less than 1 year-old cubs approach to lions – behavior independent of parents?

Minor things:

L122 subadult > subadults

L138 Fig. 1 legend, to be consistent with e.g. Supplementary Material, better to add space before and after “±”

L241 extra “this”

L390 Fig. S1 to be mentioned

Table S1 b represent misspelled

L460, L463 Add space before REFs (12) and (13)

L486 2,724 > 2724 or add coma to L455 3791

L514 I would remove “that were significant” before alpha = 0.10

L517 coma missing at the end of the sentence

L515-516 REFs missing for R. 3.6.2. and lme4 package

L581 Add REF

Supplementary Material; Table S1 L54: I guess the authors mean 166 hyenas and not 156

Table S2; Hosts of differing ages acquire infections from different sources; determinates > determinants.

In addition, hypotheses in Table S2 are lacking supporting references.

REVIEWER COMMENTS

Reviewer #1

1. **Comment:** Overall, I found the results presented very interesting (parasite manipulation is a fascinating topic not well explored in natural systems), although the manuscript deals with two main findings and thus appears a bit limited. In addition, one main finding lies on a very small sample size (N=15 cubs) and we cannot totally exclude spurious associations between minimum approach distance and infection status in cubs. However, this result based on cubs' behavior is reinforced by a second finding about the probability of dying from lions as a function of individual infection status (which I found very nice!).

I leave thus the editor deciding whether the results presented deserve publication in Nat Comm. I only few more or less minor comments (organized by line number).

Response: Thank you for your thoughtful comments and suggestions. We are glad that the reviewer appreciated the value of investigating links between putative host-manipulators, host behavior, and host fitness in a natural system. We agree that larger sample sizes could reinforce our key findings and will add value future studies. We also want to emphasize the difficulty of obtaining long-term behavioral data from free-living mammals. We have undertaken careful analyses to link behavioral phenotypes, disease status, and fitness for 168 hyenas obtained over a 30-year period. We therefore feel the outcomes represent a significant contribution to existing literature on this topic (which overwhelmingly involves even smaller sample sizes), and thus merits publication to promote follow-up work by other groups (e.g. studies focused solely on young individuals). Below, we catalogue our responses to each reviewer comment. We hope you will agree that these edits have improved the quality and clarity of the manuscript, and we are grateful for the thoughtful criticism that facilitated them.

2. **Comment:** L87: did you re-run your analyses excluding those samples with an uncertain diagnose? I think it would be worth.

Response: To address this pertinent question, we have re-run our analyses and confirmed our previous findings. Our central results (concerning cubs) were unaffected as there were no uncertain diagnoses among cubs. Excluding samples with "doubtful" or uncertain diagnoses from the subadult and adult models also did not change the direction or significance of the estimate, which was 0.27 (95% CI: -0.19, 0.72) when including all samples, and 0.02 (95% CI: -0.87, 0.92) after excluding individuals with uncertain diagnoses. We have updated the text to reflect these changes.

Results:

Pg 7, Lines 139-141:

Similarly, excluding subadult and adult hyenas with a "doubtful" *T. gondii* diagnosis did not materially change the results (0.02 [95% CI: -0.87, 0.92] related to minimum approach distance for infected vs. uninfected).

Methods:

Pg 31 Lines 624-627:

We also modeled the hyena approach distance from lions as function of *T. gondii* infection among hyenas diagnosed as either positive or negative but excluding the doubtful diagnosis category. This second sensitivity analysis aimed to rule out any potential variable misclassification bias.

3. **Comment:** -L90: the “r” coefficient is quite low (% of variance explained: 49%), why is that?

Response: The correlation coefficient is 0.70 between ELISA and IFAT assay results run on 60 hyenas. We are not sure if we are referring to the same result as the reviewer, but these results were on line 90 of the original draft manuscript. As shown in figure S2 (and discussed in next comment), only two outliers deviated from a clear correlation between the two methods. We were encouraged to see that this correlation approximates a linear fit with homogenous variance around a positive slope (see Fig S2). We found it unsurprising that the correlation is not closer to 1 given that a) the two methods employ different approaches to measuring antibodies and b) IFA values are not continuous, because plasma is tested by serial dilution until fluorescence is no longer observable under microscope. Many published studies involving *T. gondii*-infected mammals did not cross-validate ELISA results at all. Thus, we considered this supplemental cross-check (with a subset of samples) to be a good validation of our ELISA results and a strength of our study.

4. **Comment:** Also, did you exclude the obvious outlier from your sample set (Figure S2)?

Response: Yes, as noted by the reviewer, we removed two outliers from our data set for the final analyses, though we note that inclusion of these data points did not substantively change our results. This is noted in the text (Methods: Pg 24, Lines 451-455).

5. **Comment:** -L95: I found somehow unexpected that dominance rank did not influence *T. gondii* prevalence because I would have (perhaps naively) expected a relationship between boldness and dominance. Is it not the case in your study system that more dominant individuals are also more bold?

Response: Thank you for this comment. We also predicted that dominance rank would be associated with *T. gondii* infection prevalence, and that higher-ranking individuals, which have greater access to potentially infected food resources, would have a higher probability of being infected. However, this was not supported by our data, c.f. Table 1. We did find that higher ranking hyenas approach closer to lions (Table S1), supporting the notion that higher ranking hyenas are bolder. Our results replicate previous findings in which higher ranking hyenas approach closer to lions (Yoshida *et al.* 2016. *Behaviour*. 153: 1665-17722). In light of these findings, it appears that, although higher rank does

not lead to greater odds of being infected, both infection and higher rank are associated with bolder behaviors, at least with respect to approaching lions. One possibility that occurs to us is that lower-ranking individuals may be driven to eat a larger diversity and number of lower-quality prey (e.g. rodents, lagomorphs, etc.) compared to higher ranked individuals. Dominance may therefore have countervailing impacts on disease exposure and incidence. Testing this would require more nuanced analyses than our sample size and phenotype profiles permit. We chose to leave it out as we find some publications on the topic of host manipulation over-reach with data interpretation, and also crowd too many ideas into single manuscripts to effectively (incrementally) advance this important research area.

6. **Comment:** -L100: I'm wondering whether authors should present only the results from adjusted analyses. Unadjusted analyses do not bring extra information.

Response: We agree that streamlining data presentation is always valuable where possible. Here, we have chosen to provide unadjusted associations in **Tables 1 and 2** to present raw associations in the data, upon which the reader may assess the impact of subsequent adjustment for covariates. Such an approach is used in epidemiologic analyses as a valuable standalone approach for understanding crude associations (Conroy & Murray. 2020. *Brit. J. Canc.* 123:1351-1352), as well as in the context of multivariable analyses (e.g., Laubach *et al.* 2021. *Proc. Royal Soc. B.* 288: 2020815) to promote transparency in statistical analyses.

7. **Comment:** -L120: to increase sample size in cubs, I'd suggest performing analyses using sliding age-windows. For example, authors could analyze individuals aged 18 months or less; 24 months or less.

Response: We agree that a limitation of this study is the small sample size, particularly in cubs – and acknowledge this in the manuscript. However, creating modified age windows is not ideal for two reasons. First, the cub life stage ends at ~12 months based on key life history traits, like weaning, that are established in the literature (Holekamp and Smale. 1998. *Biosci.* 48: 997-1005). Second, we noted clear differences in the prevalence of infection between cubs vs. subadults and adults as well as in the minimum approach distance from lions between cubs vs. subadults and adults (see Table 1 and Table S1), indicating that altering the threshold for life stages would not be appropriate. Given the results from our bivariate analyses, and the fact that spotted hyenas exhibit discrete (vs. continuous) changes in behavior, morphology, and physiology when they transition through life history stages, we feel that the preexisting age cut-offs are well supported.

8. **Comment:** -L386: is it possible to quantify the intensity of infection? It would be interesting to correlate this quantitative measurement with boldness.

Response: This is an extremely interesting question that gets into very difficult territory

in terms of analysis/interpretation. Although categorical parasite diagnostics (e.g. negative/positive) are well-validated in the parasitology literature, other writers also raise valid concerns about interpretation of continuous variables (e.g. ELISA SP ratios) within diagnostic categories. The ELISA value of infected hosts, for instance, represent the outcome of complex interplay between the parasite and host. Thus, an infected animal with more circulating antibodies (i.e. higher ELISA) need not have a higher parasite load. Additionally, even animals with higher parasite loads are not necessarily subject to higher parasite burdens (e.g. induced maladaptive behavior and/or survival rate declines). We therefore leave this fascinating question until such time as parasite load and burden can be jointly established in the focal animals. This may require a laboratory system and/or a host (e.g. rodent) that could be sacrificed and dissected for histological or genetic analysis of parasite densities and distributions (among tissues) once field data and immunology data are in hand.

9. **Comment:** -L495: 'all other causes of death' did not include those 'ambiguous' deaths that could have been attributed to lions, right?

Response: Correct, we only included mortality data where the cause of death was known. While this reduced our sample size it is a more accurate measure of causes of mortality and therefore a conservative estimate of links to *T. gondii* infection.

10. **Comment:** -L554: I don't understand why, in the subadult-adult model, authors did not consider age as a continuous variable. I think it is important to control for age in months as done for cubs, especially because infection status depends on age.

Response: Thank you for making this point. We controlled for age as categorical variable in the subadult and adult models because for some adult animals (e.g. females who we began studying as adults and immigrant males from non-study clans) we do not know their exact birth dates. For these animals, we are unable to calculate their precise age in months. If we controlled for age (in months) we would have missing data for a number of hyenas, so we used age categories at the time of diagnosis and at the time of interactions with lions in our models. We do not think that this is problematic given that the odds of *T. gondii* infection (Table 1) and the average minimum approach distance from lions (Table S1) are similar between subadult and adult hyenas. We have added text describing our rationale.

Methods:

Pgs 30-31, Lines 608-1

Nota bene: in the subadult and adult model age was not parametrized as a continuous measure (e.g. age in months) because for some adult female hyenas, who we began observing as adults, and for some immigrant males, whose natal clan is not known, we do not know the exact birth date of these hyenas.

Reviewer #2

1. **Comment:** Gering et al., take a novel approach, by using data from a wildlife system, to test the general ecological hypothesis that certain pathogens have evolved to cause 'risky' behavior in their host species to promote transmission. Their findings suggest that hyenas infected with *T. gondii* show more risky or bold behavior, and that this behavior may drive transmission of the pathogen from intermediate to definitive host. For example, they show that infected hyena cubs approach lions more closely and are more likely to die by being killed by a lion. This is a fascinating topic and it's exciting to see these hypotheses tested in a wildlife system. The authors do an excellent job of formulating a number of interesting hypotheses and providing a detailed discussion of possible mechanisms underlying their findings. However, I had some concerns regarding both the statistical and diagnostic approaches (e.g. determination of an infected/uninfected status in individual hyenas) that need to be resolved in order to have confidence in the authors reported results/findings.

Response: We thank the reviewer for their interest in our work and the detailed feedback that they have provided. We took care with our initial analyses and appreciated the opportunity to further validate results via all reviewers' helpful criticisms. We address Reviewer #2s' comments below.

2. **Comment:** Regarding infection status, older animals may be infected, but titers may have declined below detection, so even if they test seronegative, they may have been exposed in the past.

Response: The literature on *T. gondii* infections suggests that in the vast majority of cases infected individuals retain lifelong immunity and seropositivity (e.g., as recently reviewed by Ybanez et al. 2020, and in Tenter et al. 2000). Opsteegh et al. (2011) have used modelling approaches to suggest recoveries might occur in wild boars (bringing IgG down to non-positive levels), but their SIR model incorporating this 'drop-out' only improved model fit (vs. SI model) among the very oldest individuals in their study. We do not think this scenario applies to hyenas because we see positivity rates (i.e., prevalence) *increase* (rather than decrease) with age in our dataset, such that most adults were seropositive. ELISA SP ratios (i.e., circulating IgGs) of the oldest seropositive animals in our dataset also did not dip toward the diagnostic cutoff, as shown in newly-added supplemental plot of ELISA values vs. continuous age (Figure S3).

Ybañez, R.H.D., Ybañez, A.P. and Nishikawa, Y., 2020. Review on the current trends of toxoplasmosis serodiagnosis in humans. *Frontiers in Cellular and Infection Microbiology*, 10.

Opsteegh, M., Swart, A., Fonville, M., Dekkers, L. and Van Der Giessen, J., 2011. Age-related *Toxoplasma gondii* seroprevalence in Dutch wild boar inconsistent with lifelong persistence of antibodies. *PLoS One*, 6(1), p.e16240.

Tenter, A.M., Heckeroth, A.R. and Weiss, L.M., 2000. *Toxoplasma gondii*: from animals to humans. *International journal for parasitology*, 30(12-13), pp.1217-1258.

- Comment:** The authors could assess whether the young animals always had relatively high titers, and if so, one could potentially assume that titer decline is slow enough that if an animal is < a certain age, then the failure to detect a titer means that the animal was never exposed (vs. exposed a long time ago and the titer has declined below detection). Alternatively, the authors could look to see whether there are sufficient data from known positive animals through time that a titer decay rate may be estimated. This could then be used to estimate the time that it takes a titer to decline below detection after initial exposure and only animals younger than that could be included in the analyses to increase the probability that seronegative animals are truly uninfected.

Response: We believe that a within-individual time series is the only convincing way to explore titer changes over time in this system, and our dataset does not permit this. Still, our newly added supplemental plot of age (in months) vs. SP ratio (from ELISAs) suggests that IgG levels remain high into late life. If titer drop-out were at play, we would have expected greater variance in the SP-ratios among the adult *T.gondii*+ hyenas in our dataset (but see the recently added supplemental figure, Figure S3), because these animals would collectively represent a mixture of recently-acquired infections and older infections acquired early in life.

- Comment:** Statistical approaches – rather than performing multiple ‘univariable’ or bivariate analyses, I would strongly recommend including all potential explanatory variables in the model and using some sort of model selection technique to arrive at the best fit model.

Response: We understand the rationale for this recommendation, but have also recently reviewed this topic in detail ahead of formulating our modelling approach (Laubach *et al.* 2021. *Proc. Royal Soc. B.* 288: 2020815). In brief, automated model selection techniques are appropriate when the goal of the research is prediction. Instead, we sought to test the hypothesis that boldness behaviors are influenced by *T. gondii* infection, which requires careful consideration of the temporal and causal structure among all independent variables. Statistically significant associations of independent variables and model fit are not sufficient justification to include a variable in models to test our hypotheses, as this can introduce bias in the estimates of interest and lead to inappropriate conclusions.

Comment cnt’d: I was confused as to why age was used as a categorical variable in some analyses and as a continuous variable in others.

Response: Thank you for noting this point. We have addressed this issue in response to Reviewer 1’s comment 11 (see above). In addition, we were most interested in

categorical age, as transitioning from cub->subadult->adult corresponds to biologically significant life-history stages with distinctive behavioral phenotypes. We also observed that prevalence of infection was lower among cubs and similarly higher among subadults and adults.

Comment cnt'd: I also think (but certainly the authors should consult with a biostatistician on this – as with other statistical issues) that there is no need to perform separate analyses on cubs vs. the older age classes. I believe that this would be captured by a single mixed effects model that includes age*infection status interactions and individual ID as the random effect.

Response: We are not certain which models the reviewer is referring to here (i.e., is it the model where *T. gondii* infection, minimum approach distance from lions, or cause of mortality is the outcome?). We address them separately below:

- First, in models where either *T. gondii* infection or mortality is the outcome, each hyena has a single measurement in our data set, so a random intercept for individual ID is not possible. We included a random intercept in our model of approach distance from lions in which individual hyenas have repeated interactions with lions.
- We used separate models for cubs vs subadults and adults for two reasons. Firstly, we knew, prior to beginning this work, that lion approach distances and behaviors are very different for cubs than older counterparts. Next, we further observed that (as expected), the nature of the relationship between *T.gondii* infection and approach distance from lions depends on a hyena's age within our dataset (see our response to Reviewer #1 Comment #7 and Table 1 + Table S1). In such a scenario, the appropriate approach is to conduct stratified analysis within categories of the effect modifier (see, Laubach *et al.* 2021. *Proc. Royal Soc. B.* 288: 2020815). In principle, adding this suggested interaction term (age class x infection) would test for the effect modification we observed by fitting separate age stratified models. In stratifying by age group, we isolate the effect *T. gondii* infection on a hyena's approach distance from lions, while allowing this effect to differ across age groups.

Comment cnt'd: Statistical language – throughout the MS the language describing the statistical approaches struck me as unconventional. I would strongly recommend including a biostatistician as a coauthor so that they can help write the statistical sections and advise on the statistical approaches. E.g. the authors refer to a sensitivity analysis, but this seemed to be an analysis of the contribution of other possible explanatory variables.

Response: Throughout the manuscript we have double checked and attempted to clarify descriptions of our data analysis and modeling (see, e.g., our response to the criticism immediately above). Regarding the specific example of 'sensitivity analysis,' this term and method is commonly used and appropriately described in our manuscript (c.f. Greenland. 1996. *Int. J. Epid.* 25: 1107-1116. Cited more than 500 times; VanderWeele and Arah. 2011. *Epid.* 22: 42-52. Cited more than 250 times). Thus, we

prefer to retain the current syntax unless the Editorial team is in consensus about this point.

5. **Comment:** Livestock density and human disturbance are used interchangeably, I would suggest choosing one, defining it and then staying consistent throughout the MS as to how this variable is referenced.

Response: This was a very helpful suggestion given that the two terms imply nested sources of ecological selection on parasites and hosts. We now use livestock density throughout to enhance the specificity and clarity of the article.

6. **Comment:** H1 this is an interesting hypothesis, but I'm not entirely convinced of the mechanism. I think the biological mechanism underlying this should be more clearly explained. As it stands, it doesn't make a ton of sense to me unless the hyena are predated livestock and there's a high prevalence of *T. gondii* infection in livestock, or if domestic cats are accompanying these illegal livestock grazing events (which I don't think they are...).

Response: In our study population, we observe livestock depredation by hyenas, particularly when cattle, sheep, and goats graze illegally inside Reserve boundaries. The *T. gondii* literature also supports high prevalence of *T. gondii* in livestock farmed where human-disturbed and wild areas meet. In addition, through direct observation and GPS collars, we know that hyenas are not confined to the Reserve, and they have the potential to come in contact with oocysts in the soil and water in nearby Masai villages. These oocysts can be generated by any felid (wild, feral, or domestic) that preys on intermediate hosts in areas of heavy livestock grazing –whether those infected hosts are themselves livestock, or commensal with pastoralists (e.g. house mice and rats). All of these scenarios add weight to H1, and we have added text with supporting citations (see below) to clarify this:

Discussion:

Pg 11, Lines 199-202:

We initially predicted that hyenas, through ingestion of contaminated water or consumption of infected meat, would have a higher prevalence of *T. gondii* infection when living in close proximity to domesticated animals and human commensals since these are known to serve as parasite reservoirs (J. Dubey, 2010; J. P. Dubey et al., 1995).

7. **Comment:** The authors repeatedly mention adjusted vs. unadjusted and I found this very confusing. What these two terms refer to should be stated more clearly. I believe they mean models including just the explanatory variable of primary interest vs. those which include all possible explanatory variables.

Response: We have added to the text to clarify the distinction between unadjusted and adjusted models (see below). In general, unadjusted models refer to simple regression

with one dependent and one independent variable. The adjusted models vary in their exact parameterization based upon the specific outcome and explanatory variable of interest. To make this clear, we also now explicitly indicate which additional variables are included in adjusted models in all table and figure legends (as well as in the Methods).

Methods:

Pg 29 Lines 570-574:

In addition, we also explored associations of other key demographic characteristics as determinants of infection, namely sex, age at diagnosis and social dominance rank. Following the simple regression models that contained one single explanatory variable (unadjusted analysis), we also examined multiple-variable (mutually-adjusted) associations among the above variables.

Pg 29, Lines 578-582:

To investigate the extent to which infection status is related to boldness behaviors, we used simple (unadjusted) and multiple-variable (adjusted) linear regression models in which *T. gondii* diagnosis (infected vs. uninfected) was the explanatory variable of interest, and the hyenas' square root transformed minimum approach distance (m) was the outcome. We transformed the distances to improve assumptions of normality.

- 8. Comment:** The authors mention using a restricted vs. unrestricted data set for the analyses based on the plasma *T. gondii* ELISA test results. This is fantastic! I would suggest that only the data set that they refer to as 'restricted' should be used. Or at least limit the length of a time interval between negative test results and behavioral observations. Given the high seroprevalence of anti-*T. gondii* antibodies in the hyena population and the fact that probability of infection increases with age, it's very possible that new infections could occur if the time interval between plasma collection and behavioral observation is long.

Response: We greatly appreciated the recognition of the merit of the restricted dataset. This isn't a feature of other related studies we reviewed during the design of this work. We prefer to report results from the full data set given the ease of biological interpretation, increased statistical power, thus more stable estimates, and considering that results do not materially change when the restricted data set is used. The consistency in our result between these two data sets is expected considering we had no reason to believe that there was systematic misdiagnosis with respect to our outcome, minimum approach distance from lions. Therefore, the worst-case scenario is non-differential bias (i.e. misclassifying approach distances for infected vs. uninfected hyenas) that could cause estimates trend towards the null. Thus, our results represent conservative estimate of the effect of *T. gondii* on hyena behavior.

Comment cont'd: If the authors wanted to consider including behavioral observations from a time period prior to a positive test result, one approach would be to consider the

actual titer magnitude, and if available, calculate titer decline in their hyena population using data from animals for which multiple positive test results are available. If animals have a very high titer, this is strongly suggestive of a recent infection. If animals have a low titer and both the rate of titer decline and the expected maximum titer for that population can be estimated, then it may be possible to back calculate the time of exposure (although this is a very tricky process and there may not be sufficient data to attempt this).

Response: We very much liked this suggestion; it outlines a creative way to reach beyond some of the limitations of our dataset. Unfortunately, we have low confidence in two requirements of this approach. First, we doubt that titers decline at similar rates in animals with varying health (i.e., condition) and/or immunogenetic programming. Exploring whether or not this is the case in our system, where titers are only available for a single timepoint for any focal animal, would require massive increases in both sample size and (e.g., condition-dependent trait) phenotyping. Second, we do not have repeated measures of titers for the animals in this dataset, which unfortunately make the suggested approach unfeasible.

As a more general response to the reviewers: We are enthusiastic to see so many creative ideas for further analyses/follow-ups; inspiring such is our greatest hope for the submitted manuscript. Some of these recommendations are similar to ideas we had already considered. Others are new to us, but we find them all to be logical and innovative (though some are better-suited to other model systems). For the present manuscript, we elected to err on the side of caution and avoid over-reaching with data analyses. This leaves several questions for other investigators/future studies to tackle. Our rationale for this deliberate choice was to avoid marching a reader through overly ornate data filtering/model fitting, as that approach risks diluting a clear central message. We envision subsequent, more detailed follow-up work (introducing additional methods and datasets) that could be placed into more discipline-specific journals.

9. **Comment:** For the 'death by lion' analyses – I think the authors could use a similar glmm approach as outlined above but using a logistic regression instead of a linear regression with death by lion vs. death from other cause as the outcome variable. In this case as well, I think it would be important to take into consideration the time between death and the negative test result as these animals could have become infected in the interim.

Response: First, we did use a generalized linear model. In the analysis of death by lion, the outcome is dichotomous (death by lion vs. other known sources) and so the model was a logistic model (which is a generalized linear model). Second, given that we do not have repeated data on cause of mortality from individuals, it does not make sense to use a mixed model. Additionally, thank you for raising the point about the time between diagnosis and death. Unfortunately, we do not have the data necessary to know if hyenas became infected after diagnosis and prior to death. However, hyenas diagnosed

as uninfected that later seroconverted would weaken the observed (positive) correlation between infection and mortality by lions. Thus, misdiagnosis in this scenario will lead to a conservative estimate of the effect of *T. gondii* on hyena mortality by lions, and at most increase the type II error. As mentioned in our response to the preceding comment, we contend with this type II error risk because mitigating it (e.g. via the creative proposal of the reviewer) would require data we do not possess.

10. **Comment:** Some people will analyze data using multiple cutoff values and run two sets of analyses, one which includes the ‘indeterminant result’ animals and one which excludes them from the positive ‘bin’. The authors may want to consider running such analyses to assess whether altering the cutoff to include the lower SP ratio animals changes their findings.

Response: Thank you for this helpful comment which was also raised by Reviewer 1. Our response is therefore detailed above (in our response to Reviewer 1, Comment 2). Briefly, we conducted additional analyses that removed the “doubtful” (i.e. indeterminant) diagnosis samples from the model and we observed that these results were consistent with the original analysis.

11. **Comment:** Fig. S1. This is a great figure and presents the data clearly; however I found the descriptions in the text part to be a bit confusing. You may want to consider stating “Distribution of *T. gondii* ELISA results (SP ratios) from spotted hyena plasma samples collected from animals in or near”. It may also be a bit clearer to elaborate by saying something like “The dashed red line is the upper SP ratio cutoff for negative diagnoses, i.e. ratios < XX are considered negative, and the solid redline is the lower cutoff for positive diagnosis, i.e. all ratios > 0.5 were considered positive.” You could also simplify by just say something like “SP ratios below the dashed red line were considered negative, those above the solid line were considered positive and those between the lines were indeterminant”.

Response: We found these suggestions helpful and have incorporated them into the legend as advised (see below):

Supplementary Materials

Figure S1, Legend:

Fig. S1. Distribution *T. gondii* ELISA results (SP ratios) from spotted hyenas sampled in or near Kenya’s Masai Mara National Reserve. The dashed red line is the upper SP ratio cutoff for negative diagnoses (SP ratio ≤ 0.40), and the solid red line is the lower cutoff for positive diagnosis (SP ratio ≥ 0.50). The region between the red lines corresponds to the SP ratio range for “doubtful,” which were treated as negative in this study.

12. **Comment:** Fig. S2. Please include what is considered a positive result for the IFAT. This is important as the IFAT results seem to indicate that there was detectable fluorescence at some dilution in all samples. If the cutoff between positive and negative is fluorescence

at any dilution vs. no fluorescence, then it seems that all of these samples are positive. This would obviously make any subsequent analyses impossible. Also, I'm not sure if this is the right analysis to perform in order to assess comparability between tests. A Cohen's Kappa (McHugh ML. Interrater reliability: the kappa statistic. *Biochem Med (Zagreb)*. 2012;22(3):276-282) may be more appropriate with test results binned into positive and negative categories, but I would consult with a biostatistician to confirm this. If the correlation is the correct test to use, I'm still not sure that a correlation of 0.7 indicates good agreement, but again, best to consult with a statistician about this. There are a few other diagnostic test issues to consider: for the ELISA, is the positive control a hyena positive control? I believe that the IFAT is generally considered to be the diagnostic test of choice for determining serum (or plasma) antibody titers, so it would be good to be able to provide strong evidence that the ELISA results are roughly equivalent. But I also found this article where the authors found good agreement between an ELISA and IFAT for detecting anti-*T. gondii* antibodies (Glor, S.B., Edelhofer, R., Grimm, F. et al. Evaluation of a commercial ELISA kit for detection of antibodies against *Toxoplasma gondii* in serum, plasma and meat juice from experimentally and naturally infected sheep. *Parasites Vectors* 6, 85 (2013). <https://doi.org/10.1186/1756-3305-6-85>). They also used the Kappa to assess agreement between tests, so perhaps a good reference.

Response: IFA is only considered to be positive if fluorescence persists beyond a system-specific (validated) cutoff. When this value is not available from reference panels (e.g. in spotted hyenas), some investigators have used workarounds. We find these workarounds somewhat unconvincing, whereas the ELISA cut-offs have been cross-validated in a variety of host species. As mentioned above in our response to reviewer 1, our only purpose in presenting the IFA data is to see whether (or not) positive relationships between IFA and ELISA values corroborated the general output of the ELISA assay (since we do not have a reference panel of known infecteds). We therefore present r , rather than Kappa, because we do not have binary responses for IFA in this system. In other words, the measured differences in IFA cutoffs provide an index of relative circulating antibody levels but not a diagnostic. We are also aware that interpretation of r is somewhat confounded by the fact that IFA levels can only occur at intervals along the dilution series. While an advanced statistical pipeline could be developed here, we really just wanted to show the reader that 1) an ancillary investigation confirms the ELISA plate is working as designed, and 2) there are neither glaring non-linearities nor variance heterogeneity in the ELISA and IFA datasets (a finding that is clearly visible in the plot). This assured us that cross-reactivity and/or plate saturation did not compromise the adaptation of the multi-species ELISA workflow to our hyena plasma analyses.

13. **Comment:** Table S1. I would suggest reporting the SD vs the SE so that the reader can more readily assess the variation in the data. I found the superscript descriptions confusing, I would suggest working to clarify these. For example, instead of "From an independent t-test for sex, food presence, and human disturbance; from a Wald chi-

squared test for age group” perhaps have one superscript indicating that the analysis/p-value was from an independent t-test, and then mark the appropriate corresponding values, similarly for the Wald chi-square.

Response: We agree with this suggestion. In Table S1, we report SE because the estimates came from a linear mixed model that accounted for multiple measurements of observed minimum approach distance to lions for some individuals. We have changed the footnotes and superscripts as suggested by the reviewer in Table S1.

14. **Comment:** Food presence and whether the lion was an adult male were not included in this table, I would suggest including these even if they are not significant predictors of distance. Otherwise, it’s confusing that these explanatory variables are mentioned in the superscripts but are not present in the table.

Response: Thank you for noting this. We have updated the footnotes and the title of this table to indicate that we report estimates from bivariate models assessing associations between potential confounding variables with hyenas’ minimum approach distance from lions in this table.

15. **Comment:** There were typos and grammatical errors in the C superscript: I think it should be “cubs’ maternal...” also interaction, not interact.

Response: Thank you for catching this typo. We have changed the text.

16. **Comment:** Also, a mean of -5.14 for Dominance Rank doesn’t make sense. Do the authors mean a reduction in the mean minimum approach distance of 5.14 with each 1 unit increment in standardized rank? If so, please adjust accordingly. Or perhaps I’m not understanding what’s being reported in this table?

Response: Yes, this is the correct interpretation as noted in the table under the Dominance rank is ‘Per 1 unit increment in standardized rank’.

17. **Comment:** Movie S1. This is a really interesting video! I’m wondering though whether this sort of interaction would really represent a significant transmission risk. Infection through blood seems a very unlikely scenario. Toxo encysts in muscle and neural tissue and these tissues would be the most likely source of infection during a lethal lion-hyena interaction. However, if lions aren’t eating the hyenas either, it seems that transmission through consumption of muscle or neural tissue is also unlikely. But I’m not a *T. gondii* expert, so perhaps best to consult with one. If this is a likely transmission scenario, then I would recommend including a reference as to support the fact that transmission can occur through these routes and/or through ingestion of even a very small amount of infectious material.

Response: Thank you. We have added a reference (Siegel *et al.* 1971. *Blood*. 37: 388-394) in the Discussion that provides evidence of the transmission of toxoplasmosis via leukocytes. The parasite can migrate within hosts via blood, especially during acute infection stages. Tachyzoites in muscle tissues could also be readily ingested through the killing event seen in the film. We acknowledge that all of these factors together may still comprise limited transmission opportunities for the parasite. That is the primary reason why we elaborate on non-adaptive (collateral) manipulation scenarios within the Discussion, and state that no scenario can be definitively supported or excluded by a simple correlation of infection, host behavior, and host fitness.

18. **Comment:** Results: I'm concerned that given the very high SP of anti-T. gondii antibodies that the adults that are negative are not true negatives, but rather animals with titers that have dropped below detection. An assessment of antibody titer changes through time in single individuals could help provide insights into whether this is an issue or not. I.e. if titers remain at fairly constant levels post exposure, then this wouldn't be an issue, but if T. gondii titers decline through time, since we know the protozoa isn't cleared, this would suggest that some of the apparently negative animals (especially those falling within the 'indeterminant SP ratio range) may actually just be animals that are infected with T. gondii, but that infection occurred long enough ago that antibodies are no longer detectable. One thing that could be done with the current data would be to assess whether the SP ratio correlates with age. Presumably, animals with more recent infections will have higher titers. So, if infection risk increases with age, then younger positive animals are likely to have been exposed more recently than older positive animals. Therefore these younger animals may have higher titers, while older animals that may have been infected many months or years prior, may have lower titers. If the authors find this to be the case, then they should consider the possibility that the test negative older animals may in fact be infected but seronegative. It is also worth doing a literature review to assess whether there is evidence of this phenomena occurring (i.e. seronegative animals that are confirmed positive after necropsy and PCR or special staining of tissues).

Response: We respond to this concern above including citations (see Reviewer 1, comment 2). In brief, we do not believe older animals are 'aging out' of seropositivity based on 1) literature suggesting infected hosts retain lifelong immunity and seropositivity, 2) the observation that prevalence is much higher in older vs. younger age categories, and 3) the observation that older animals with positive SP are not closer to the SP-cutoff for positive diagnoses than younger counterparts.

19. **Comment:** Table 1. Typos in superscript d: I think it should be cub's maternal (not cubs matnema!)

Response: Thank you. Done.

20. **Comment:** Figure 1. The mean +/- SD would be more informative, I would recommend changing accordingly.

Response: We agree that showing estimated means and raw variation (i.e. SD) is useful. We have modified Fig. 1 to include mean and SD for cubs in panel A. As noted previously, for subadults and adults there are repeated distance measures per hyena. Therefore, we report the marginal means +/- the SE from a mixed model that includes *T. gondii* infection status as an explanatory variable, distance as a continuous outcome, and a random intercept for hyena ID in panel B.

21. **Comment:** I also don't understand why the hyena age group on the date of diagnosis would be relevant, assuming that this date occurred prior to the date of the hyena-lion interaction. I would think only the age of the animal at the time of interaction would be relevant.

Response: Because the prevalence of *T. gondii* infection is age structured, with older hyenas having a higher probability of being infected, we controlled for hyena age at the time of diagnosis. Adjusting for such a variable (known as a precision covariate) increases model efficiency and power (c.f. Schisterman *et al.* 2009. *Epid.* 20: 488-495.)

22. **Comment:** Table 2. Tables should be able to stand alone without looking at the text. I would suggest clarifying what is meant by adjusted and unadjusted, as written in the table, it's very confusing. In the MS text (Lines 123-126) it is more clear.

Response: The adjusted/unadjusted model descriptions were also criticized by Reviewer comment #2. Our response outlines steps taken to clarify how adjustments were made and what purpose they serve. In addition, we explicitly state in the footnotes which variables are controlled for in our adjusted models

23. **Comment:** Lines 127-135: I believe that a sensitivity analysis is when you look at how variation in a given variable impacts output, not how addition of new variables impacts output. I think what the authors are doing here is assessing the significance of various additional potential predictor/independent variables on the dependent or outcome variable (i.e. minimum distance to lion). If this is the case, then I think a more appropriate approach would be to determine the best fit model by including all potential predictor variables in the glm and then use a model selection process to determine the best fit model. I was also a bit confused by the wording of the last sentence. I think the authors mean that the model estimated the approach distance as a function of *T. gondii* infection status (as determined by serostatus), age, sex etc., this is not clear as currently worded.

Response: A sensitivity analysis is broadly defined as any extra analysis that tests the robustness of study findings, whether that be through the addition of covariates, changes to how covariates are parameterized, omission of certain individuals who may

be different from the majority of study population (e.g., possible outliers) etc. Thus, what we describe as sensitivity analyses are indeed sensitivity analyses, as we pointed out in response to the reviewer's previous comment 4. We understand that the confusion comes from a narrower colloquial use of the term within EEB, but have checked the definition in several credible sources and are confident this does pertain to the described analyses.

24. **Comment:** Lines 150-151: I think a chi-square or Fisher's exact should be used to assess the statistical significance of this difference between infected and not infected (i.e. 52% vs. 25% died from lions vs. other causes). Or, as I suggested above, a logistic regression could be used for these analyses.

Response: We used a logistic regression as the reviewer suggests.

25. **Comment:** lines 153-158: I found this a bit confusing. I wasn't sure what the relationship between the statement in lines 155-156 was to the last sentence of this paragraph. Please clarify.

Response: We have simplified this text to enhance its clarity as quoted below.

Results:

Pg 9, Lines 168-170:

In this small subsample of 11 hyenas infected as cubs, all of which were sampled between 1990-1999, the probability of dying by lions vs. other known sources of mortality was greater among infected than uninfected individuals (Fisher's Exact Test $P=0.01$).

26. **Comment:** Table 3. Same question as mentioned above regarding the use of the term 'adjusted'. Please clarify what is meant by 'model is adjusted for sex'.

Response: Adjusting for a variable means that we have included it as a covariate in the model. Thus, a model that is "adjusted for sex" includes sex as a covariate.

27. **Comment:** Lines 168-171: an alternative hypothesis may be that essentially all subadult and adult hyenas have been infected, but some no longer have detectable titers.

Response: As described above in responses to Reviewer 1 and 2, the general consensus is that infected animals remain IgG seropositive for life, and (for reasons outlined above) we do not believe our system applies to the very limited putative examples of seroconversion to negativity in *T. gondii*-infected hosts.

28. **Comment:** Methods: Minimum approach distance to lions: The authors included data if hyena's distance from lion is >75 m and they include it as 75 m. It seems that this could significantly alter the calculated mean depending on the true value. Perhaps it would be

best to bin all animals $100 > X > 75$ (regardless of whether the exact number is known or not)?

Response: Among the hyena distances from lions that were recorded in the field as inequality (e.g. $> 75\text{m}$) of which there were 72 in our sample, only 5 were retained in our analyses. We conservatively set these distances to 75m so that these values would, like the rest of our 2,724 approach distances, be a numeric measure that could be entered into a linear mixed model. Binning our distance measures a categorical variable would result in a massive loss of information.

29. **Comment:** Death by lion: How did the authors deal with deaths due to unknown causes, i.e. could have been a lion but the carcass was too damaged to determine?

Response: We only included in our analyses cases in which the cause of death was known. As shown immediately below, we added additional text to clarify this point. Again, while this reduced our sample size - it is a more accurate measure of cause of mortality and therefore results in more conservative estimate of a putative association between lion-inflicted mortality and infection.

Methods:

Pg 28, Lines 544-546:

In our analysis, we dichotomized cause of mortality as death by lion vs. all other known causes of mortality and evaluated this as a binary outcome in the statistical analysis. We did not include data in which a hyena's cause of death was unknown.

30. **Comment:** STATISTICAL ANALYSES: Rather than performing individual linear regressions for each potential explanatory variable, I would suggest including all in the model and performing some sort of stepwise process to remove those that don't contribute significantly to the model fit. There are a number of ways to do this, I would recommend consulting with a biostatistician to determine the best approach for your data.

Response: We believe we have addressed this concern in response to the reviewer's comment 4.

31. **Comment:** In discussion section the authors mention additional future analyses – I would recommend that they consider some of these for this MS. Especially the ones that consider the number of other hyenas involved in the hyena – lion interaction as well as their behavior. I imagine hyena boldness would increase as the number of additional hyenas increases. Similarly, I think it is important to note the number of lions involved in the interaction. Again, I imagine as the number of lions involved in the interaction increases, the boldness of the hyenas would decrease. I don't know what the most reasonable statistical approach would be to assess this, likely some sort of hierarchical Bayesian framework (as the authors mentioned), but given all of the factors that could have a biologically important and statistically significant impact on hyena

boldness, I think it's important to consider and include all of these (or select interactions that control for this, e.g. only include single hyena, single lion interactions in the analyses).

Response: We are glad that the reviewer appreciates our suggested future directions. These ideas were included in the future directions section because they require additional data and analyses which are currently not feasible. While we agree that these additional analyses and collection of more data are warranted, they are beyond the scope of this work, which we believe is already sufficiently involved and complex. Our motivation to point out some plausible future directions is twofold. First, it is our hope that we or others can collect additional data to further test the hypotheses and alternative hypotheses addressed in this manuscript. Second, we feel that suggesting additional modeling approaches improves transparency by noting potential limitations of the current study. We hope the reviewers and editor agree that our initial work represented in this manuscript is interesting and an important contribution to the field, and that through publication it provides a next step upon which future studies can build. Lastly, reviewers have already observed that the central conclusions of this manuscript involve relatively small sample sizes (esp. number of cubs). We have already incorporated several variables into our models that we expected to be of cardinal importance in measured outcomes into our models (e.g., sex, dominance, food presence, male lion presence). The risk of any particular engagement with lions will involve extremely complex interactions (between the contested territory or food item, the hyena group size, the numbers of lions, the presence of coalition-mates, visual setting, lion condition, hyena condition and hunger, etc.). Prior analyses indicated that hyena approach to lions depends on number of hyenas present and male lion presence, although number of lions present did not affect hyena approach (Montgomery Ph. D. dissertation). However, including number of hyenas present at the time of the approach was beyond our capabilities, especially as we considered minimum approach per session, over which time the number of hyenas present may change drastically due to the fission-fusion nature of hyena societies. Introducing further covariates into the model structures risks over-fitting, but also oversimplifies the other variables that may be at play. Given this complexity, we felt safer to begin by simply asking about overarching patterns. We strongly believe this was the right approach to designing the very first test of whether (or not) *T. gondii* infection is associated with fitness-related natural behavior toward felids in free-living hosts.

Reviewer #3

1. **Comment:** This well-written manuscript describes an innovative and longitudinal study showing higher boldness in *Toxoplasma gondii* infected hyenas (intermediate carnivore/carrion host) toward one of the definitive host of the parasite in the wild; the African lion. Beside a couple of clarifications needed in the Methods and a couple of points to consider in the interpretation of the results, my main concern lies in the 'novelty' of the manuscript given the authors previous publication: "Times makes you older, parasites make you bolder – *Toxoplasma gondii* infections predict hyena boldness toward definitive lion hosts" in Banzhaf, W., Cheng, B. H. C., Deb, K., Holekamp, K. E., Lenski, R. E., Ofria, C., ... Whittaker, D. J. (Eds.). (2020). *Evolution in Action: Past, Present and Future. Genetic and Evolutionary Computation*. I understand that the nature and motives of both writings may differ but the study is essentially the same. As stated in the book chapter, the authors seem to consider the submitted manuscript as a follow-up study. If so though, it is strange to me that they do not even refer to their pilot, particularly for the serology results. In the submitted manuscript, the authors tested three hypotheses: (H1) higher livestock density is associated with higher risk of *T. gondii* infection in spotted hyenas; (H2) infected hyenas behave more boldly towards lions than uninfected hyenas, as indicated by a shorter minimum approach distance; (H3) *T. gondii* infection imposes fitness costs on the host, as indicated by greater odds of death by lion(s). In the previously published book chapter, H1 and H2 were also tested and provided similar results (for H1) and more refined ones (for H2; i.e. different models for cubs and sub-/adults). One difference may reside in the narratives. While in the former writing, the authors introduced the concept of 'collateral manipulation' – as they judged the parasite transmission unlikely to occur between hyenas and lions due to the lack of consumption by the latter – they revised their theory in the current manuscript adding the fact that blood and tissue ingested by lions may also transmit the parasite. In sum, the authors should clarify what new input this manuscript brings compared to their recently published book chapter.

Response: Thank you for carefully reviewing our manuscript. As noted, we have previously published, a non-peer reviewed, invited commentary that included some preliminary analyses on *T. gondii* infection in spotted hyenas. We have recently discussed the overlap between our current manuscript and the previous *Festschrift* with the Editor, and we copy part of that letter here. At the time we submitted our *Festschrift* manuscript, which was now several years ago, our research on the determinants and consequences of *T. gondii* infection in wild hyenas remained preliminary (as noted in the chapter and quoted below). Nonetheless, the editors of the volume encouraged us to submit preliminary analyses for inclusion in the *Festschrift*, opining that the completed study would be better suited to a high-profile journal that would a) garner attention from the scientific community proportional to the significance of the study and findings, and b) present conclusions derived from more advanced and cautious analyses of the data. These analyses now include responses to several helpful criticisms from all three reviewers outlined in the present document/response to review. Anticipating this, we included in the *Festschrift* chapter the following statement:

“The Results and Discussion provided below are somewhat preliminary; they are based on our pilot studies of the data available when an accompanying *Festschrift* volume was being assembled. As noted in the addendum, we are now compiling and analyzing a much larger dataset of spotted hyena behaviors, life history data, and disease diagnostics. The results will be presented in a forthcoming manuscript.”

As a result of our further work on the project, our recent submission to *Nature Communication* departs from the *Festschrift* chapter in several ways, including: a) refined analyses of prevalence that point to different conclusions than those reached in the *Festschrift* chapter about temporal changes in *T.gondii* prevalence, b) inclusion of additional lion hyena interaction data and sensitivity analyses that exclude several potentially confounding variables c) discussion of causal mechanisms for age-structure in infection-behavior covariates, d) and a streamlined narrative for the manuscript focusing on lion-hyena interactions. Given the novelty of the manuscript’s central results, and their potential impact on future studies, we felt this work should go through a rigorous peer review and that *Nature Communication* would be an excellent venue for the communication of our completed study. We also agree that the preliminary chapter should be cited and explained in the manuscript submitted for the present peer-review and have done so in the revised Introduction.

Introduction:

Pg 4, Lines 74-79:

While the design and objectives for this body of work were presented along with preliminary findings in a non-peer-reviewed *Festschrift* volume (Gering et al., 2020), the present manuscript includes more rigorous models that include additional candidate covariates to arrive at somewhat modified conclusions. We also present novel syntheses of our research findings in light of existing literature and point to important next steps in studying putative parasitic manipulation in wild hosts.

2. **Comment:** L54-55 Introduction. It may be necessary to remind readers how *Toxoplasma gondii* can be transmitted.

Response: We have added a sentence in the Introduction.

Introduction:

Pg 3, Lines 50-52:

Parasite transmission can occur by ingestion of *T. gondii* oocysts shed from felids (the definitive host), consumption of infected tissue (i.e., bradyzoites) from intermediate hosts, and congenital infection (J. Dubey, 1998; Elmore et al., 2010).

3. **Comment:** The Introduction would be the place to introduce the authors’ pilot study, and the novelty of the current study.

Response: We have done this according to the reviewer's suggestion (see comment #2)

- Comment:** L84 Serology results for IgG to *T. gondii*: These seem essentially the same data and findings as in the authors' previously published book chapter, regarding the number of tested individuals (168 – 2 discarded samples = 166 vs. 168 previously); the reported number of seropositive (109 vs. 110) and seronegative (36 vs. 37) individuals; as well as doubtful diagnostics (21 in both writings). The only new addition I see here is the use of IFAT diagnostics for a subset of the samples to compare with ELISA, showing high reliability. Since a larger dataset does not seem to have been used for these analyses (compared to Gering, Laubach et al. 2020), the authors should make clear what differs – if anything – in terms of results, and if not, simply refer to/cite their previous publication for this part of the results.

Response: As previously mentioned in response to the reviewer's comment 1, we believe it is important to include all results in this manuscript. The reasons include: the lack of external peer-review in the *Festschrift* and updated models (including, as noted by the reviewer, removal of outliers discordant between the ELISA and IFAT results). We chose not to refer to the *Festschrift* for results because some of our preliminary analyses (e.g. temporal shifts in disease prevalence) outlined therein were not supported after incorporating age structure into models.

- Comment:** L90 If the data (IFAT & ELISA) do not follow normality, it may be better to use Spearman rank correlation or Kendall rank correlation instead of Pearson correlation. Spearman and Kendall would rank all the values and correlate the ranks. They may also better account for potential outliers in this subset of samples that can drive correlation coefficients using Pearson.

Response: As mentioned above in responses to reviewers 1 and 2, we believe that the plot of Elisa vs IFAT and summary r-value statistic achieved what we wanted – a simple validation of the core diagnostic (here, ELISA) that is often lacking from parasitological studies. We think it highly evident from a visual inspection of the data (i.e. the plot) that a spurious correlation is not being driven by overleveraging of outliers.

- Comment:** Results. I would recommend adding subheadings that refer to the three main hypotheses (see Statistical analyses in the Methods) being tested.

Response: Thanks, this is a good idea and we have added subheadings to the results.

- Comment:** L129-130 Did the authors test whether there was a correlation between the infection status of the cubs and the one of mothers? Given the transmission of the parasite, one could predict that times of exposure to the parasite could be similar between mothers and cubs given their proximity.

Response: We are unable to test whether relatives (e.g. mother/offspring) acquire infections at similar timepoints because we do not have diagnoses for relatives at concomitant timepoints. This is an intriguing prospect, but our central aim was to test relationships between infection, behavior and fitness howsoever infections are acquired.

Comment cnt'd: In addition, how far were mothers from the cubs when a lion was in sight? We may also expect a correlation between the minimum approach distance exhibited by cubs and mothers, i.e. mothers of infected cubs getting closer to lions. This might be something to add in the discussion – whether infected cubs were on their own when exhibiting closer distances to lions or were there other individuals around. In other words, I was wondering how was the social dynamic like during these approaches. On the same line of thought, I was also wondering if infected mothers would be more permissive/less protective toward their cubs compared to uninfected mothers.

Response cnt'd: These are fascinating questions. Unfortunately, as mentioned above in our responses to Reviewer#2, we do not have sufficient data to answer them. For example, while we record all hyenas present during interactions with lions, we do not note any distances between different hyenas but only between each hyena and the lion. While maternal presence and behavior are almost certainly important, many other factors (resource presence, coalition mates, visual field, prey abundance, position on home territory, etc.) likely interact to shape lion-hyena interactions. We therefore chose to isolate any apparent effects of *T. gondii* after adjusting for a core list of context-related variables that our three decades of observation in this system suggested were both accurately measurable and of cardinal importance.

8. **Comment:** L146 Between Table 1 & Figure 1 / Table 2, it may be helpful to remind why there is a drop of $49 - 15 = 34$ cubs in the analyses. It should be added in one of the table's/figure's legend that 15 is the number of (infected/uninfected) cubs for which proximity data with lions were available.

Response: We have updated the figure legend to address this helpful suggestion. As clarified in the revised manuscript, only 15 cubs were analyzed vis-a-vis distance from lions because these were the only animals that were both diagnosed as cubs and observed interacting with lions during the cub life stage.

Results:

Pg 8, Lines 153-155, (Fig. 1 legend):

A. Among cubs (N = 15), estimates are raw means and standard deviations based on average minimum approach distance for the uninfected and infected. The cub data set includes only hyenas for which both diagnosis and distance from lions were measured as cubs.

9. **Comment:** Table 3. The difference in probability of dying from lions vs. other causes between infected and uninfected individuals could also be due to the inequality of

known causes of death between these two groups (N=25 for infected individuals and N=8 for uninfected ones).

Response: We acknowledge that the sample sizes are unbalanced. The direction of the effect trends in the expected direction (based on the host manipulation model). This inequality reflects the prevalence of infection among subadult and adult hyenas (here 75% infected vs in the larger data set 74-80% infected), indicating that this subsample of hyenas for which we have mortality data is similar to the larger data set with respect to infection. Still, as the reviewer noted, some of the unknown causes of death for uninfected individuals may be death by lion which could lead to an underestimation of the odds of death by lion for these individuals. However, given that we have no way of knowing the actual cause of death for most of our hyenas, we are making the assumption that the data are missing completely at random (i.e., that the distribution of cause of death is the same for infected vs. uninfected individuals among the hyenas for whom we do not have data on cause of death), which is common practice in studies of large observational datasets. For the sake of conservatism we used a two-tailed test, and we report that the effect is not significant. In the cubs-only analysis described after this section, we employ Fisher's exact test and thereby account for sample sizes.

10. **Comment:** Discussion. It may be interesting to reflect on how lions may want to avoid the parasite – if costly to them – by e.g. avoiding consuming infected individuals / having evolved recognition mechanisms of potential sources of infection.

Response: This is an interesting idea. Unfortunately, we do not have any data on the lion behaviors, so we cannot say much about this. In principle, *T. gondii*'s optimal strategy might be to 'take it easy' on definitive feline hosts as they can continue shedding oocysts during infection recrudescence. There are conflicting reports in the literature about the relative cost and consequence of infection to felids (vs. intermediate hosts). We find this fascinating but outside the scope of our study which acknowledges very clearly that the association we report might not be exerting significant evolutionary influence over *T. gondii*'s interaction with hyena hosts.

11. **Comment:** LL173-178 Have lions been tested positive to *T. gondii* in the Masai Mara? One of the next logical steps, as suggested above, would be to investigate whether the susceptibility of infection in lions is correlated with their killing/predatory behavior.

Response: Recent work in the Serengeti (Ferreira *et al.* 2019. *IJP: Paras. Wild.* 8: 11-117) found that 100% of lions (n=15) tested positive for *T. gondii*. We have not confirmed the prevalence of *T. gondii* in lions in the Masai Mara, but considering that our study location is part of the larger Serengeti ecosystem, we suspect a high prevalence of infection among lions that interact with our study population. Again, unfortunately we do not have any data on lion behavior at this time. We absolutely concur that this would be a fascinating research area. *T. gondii* might benefit if other genotypes are ingested by a feline host because they allow for recombination, but also

pay costs if coinfecting lineages compete for feline hosts' resources and/or impinge on these hosts' longevity.

12. **Comment:** LL227-228 The study by Poirotte et al. on *T. gondii* infected chimpanzee's attraction toward predator urine was performed with captive chimpanzees and thus the term 'wild hosts' here is confusing and inappropriate.

Response: We have corrected the wording to reflect that these chimpanzees were captive. This further highlights the novelty of our study, which is the first to document associations between infections and naturally-occurring, fitness-related behaviors between the parasite's non-definitive and definitive hosts.

Discussion:

Pg 13, Lines 239-241:

Only a small body of work examines *T. gondii*'s relationship to behavior outside of laboratory rodents and human hosts (e.g., sea otters and chimpanzees (Poirotte et al., 2016; Shapiro et al., 2019)); this work also examines a very narrow range of taxa despite the fact that a much wider array of mammals and birds are susceptible (J. Dubey, 2010).

13. **Comment:** Perspectives: How infected hyenas might be generally less responsive toward (i.e. show higher latencies to approach) a simulated threat (see Turner et al. on field experiments with a simulated hyena) is something important to reflect on and lacking from this manuscript. This previous result (Gering, Laubach et al. 2020) converge with the hypothesis that infected hyenas may be less risk sensitive, disregarding the origin of the treat.

Response: Thank you for this suggestion. We decided not to include some of the preliminary results reported in the non-reviewed *Festschrift* to provide a more focused, short format manuscript that also relies on a larger dataset (vs. the analyses of responses to simulated conspecific intruders as presented in the *Festschrift*). We would very much like to better characterize the responses of recently-diagnosed animals to a variety of experimental stimuli, but this will require development of non-invasive diagnostic methods (e.g. saliva assays). Meanwhile, we remain concerned that presenting and interpreting the underpowered conspecific intruder dataset (alongside the more robust analyses laid out in the submitted manuscript) would draw reviewer ire and induce skepticism in readers about the general credibility of the study. We were less concerned in the case of the *Festschrift* as this was meant to be an overview of a broad array of studies in various stages of development to celebrate the founder of NSF-BEACON. It may help the reviewer to learn that other chapters in this volume included personal essays about time at the BEACON institution, humorous pieces, etc.. In other words, the book was not designed to be a primary literature source so much as a multifaceted honorary tribute to Dr. Goodman upon his retirement.

14. **Comment:** L388 Methods. It would be useful to state here that one plasma blood sample was tested per hyena.

Response: Done.

Methods:

Pg 23, Lines 437-440:

We tested 168 plasma samples from 168 individual spotted hyenas and determined infection status based on the kit manufacturer's criteria for interpreting S/P: $\leq 40\%$ = negative result, $40\% < S/P < 50\%$ = doubtful result, $S/P \geq 50\%$ = positive result (**Figure S1**).

15. **Comment:** LL445-448 How many hyenas were sampled for each low and high livestock density areas? The same samples as in Gering, Laubach et al. 2020 seem to have been used although partitioning seems different – if so, the authors should mention the number of samples collected for each locality and/or refer to their previous book chapter.

Response: We have taken your earlier suggestion to cite the book chapter in the Introduction. We also double checked the sample sizes, which can be found in Table 1. In previous analyses we had hoped to be able to tease apart both spatial and temporal differences in livestock density, but in the end we only had sufficient power to consider this variable as a two level (i.e. high vs. low) factor, hence their collapse in the submitted manuscript. This could present issues with interpretation if we had found a relationship between prevalence and livestock density, but this was not the case. We therefore claim only that expected relationships were not found, but nonetheless additional research may yet reveal effects of disturbance on hyena-toxo interactions within the Mara region.

16. **Comment:** L537 Less than 1 year-old cubs approach to lions – behavior independent of parents?

Response: As mentioned in earlier responses to reviewers 1-3 above, we acknowledge that there are many credible factors (including parental behavior, group composition, ecological contexts, focal animal condition, etc.) which could impact the behaviors of our focal animals, and we say this in our Discussion. Nevertheless, we simply cannot examine all the compelling candidate covariates without over-exploiting a dataset of <170 animals once we have sacrificed power/sample sizes to account for age, sex, dominance, parental rank, etc. We do hope the very same exciting questions posed by the reviewers will be raised in our study's readers, as they provide rich fodder for further study. After acknowledging the inherent limitations of our study, we also want to reiterate that it is the first study to demonstrate covariation between *T. gondii* status and naturally-occurring, fitness-related behavior towards felines in any wild host. Howsoever these associations emerge (and we offer several credible mechanisms in our Discussion), our project is by itself a major advancement to the existing literature

because it corroborates a key prediction of host manipulation that has never been shown in the wild. Although nearly all of the flagship studies we cite within our Introduction and Discussion have been widely cited, and appear in high-profile journals, most also a) rest on thinner evidence (e.g. simpler behavioral phenotyping, no adjustments for age/dominance, no repeat measures of behavioral outputs, etc.), b) use smaller sample sizes, c) deploy less nuanced modelling, and d) involve captive/semi-captive model systems with less relevance to natural ecosystems. We believe we have already exhausted the analytical limits of the present dataset and that the results warrant communication to an interdisciplinary readership. In doing so, we prefer to avoid overreaching with both analyses and interpretation.

Minor things:

Response: Thank you for carefully noting these minor edits.

17. **Comment:** L122 subadult > subadults

Response: Done.

18. **Comment:** L138 Fig. 1 legend, to be consistent with e.g. Supplementary Material, better to add space before and after “±”

Response: Done.

19. **Comment:** L241 extra “this”

Response: Done.

20. **Comment:** L390 Fig. S1 to be mentioned

Response: Done.

21. **Comment:** Table S1 b represent misspelled

Response: Done

22. **Comment:** L460, L463 Add space before REFs (12) and (13)

Response: Done.

23. **Comment:** L486 2,724 > 2724 or add coma to L455 3791

Response: Done.

24. **Comment:** L514 I would remove “that were significant” before $\alpha = 0.10$

Response: Done.

25. **Comment:** L517 coma missing at the end of the sentence

Response: Done.

26. **Comment:** L515-516 REFs missing for R. 3.6.2. and lme4 package

Response: Done.

27. **Comment:** L581 Add REF

Response: It is not clear what should be referenced here, given this text describes what statistical test we used.

28. **Comment:** Supplementary Material; Table S1 L54: I guess the authors mean 166 hyenas and not 156

Response: Thanks for noting this, but we in fact mean 156 which is the number of hyenas for which we have data on approach distance from lions. The sample size of 166 is the total number of hyenas that we were able to diagnose for *T. gondii* infection.

29. **Comment:** Table S2; Hosts of differing ages acquire infections from different sources; determinates > determinants. In addition, hypotheses in Table S2 are lacking supporting references.

Response: Thank you for catching this typo. We have also updated the Table S2 legend to clarify that these hypotheses are potential mechanisms that we propose.

Supplemental material:

Pg 6, Table S2 legend:

Table S2. The table contains a non-exhaustive list of mechanisms that we propose might generate relationships between host age, host behavior, and infection by a behavior-altering parasite (e.g., *T. gondii*).

REVIEWERS' COMMENTS

Reviewer #1 (Remarks to the Author):

The authors have addressed most of my comments and answered the other ones. I therefore have nothing else to add. Good luck with your impressive field research station!

Reviewer #2 (Remarks to the Author):

General comment:

The authors did an excellent job of thoroughly and clearly responding to and addressing my (many) comments! Thanks for the additional thoughts and information, I learned a lot. I am satisfied with the changes in the MS. Please see below for a few minor comments/edits.

Minor comments/edits:

Regarding the animals with a 'doubtful diagnosis', rather than excluding them, two sets of analyses could be run: 1) with them categorized within the 'positive' group; 2) with them categorized within the 'negative' group.

Typos etc.

Figure S1. SP Ration.... – should be ratio and missing end parenthesis. Also, please define SP in this caption. It should stand alone without the need to reference the MS. Also, suggest indicating that distance units are in meters.

Figure S3 SPratio – should be SP ratio. Also, I would recommend staying consistent with color schemes, lack of consistency may cause considerable confusion or misinterpretation in a reader who scans or quickly reads the MS. In some figures red = positive, in others red = negative.

Table S2 “.. they might swap effects...” should this be “SWAMP effects”?

Reviewer #3 (Remarks to the Author):

I thank the authors for addressing my comments and questions regarding their manuscript - as well as my concern regarding their previous (non peer-reviewed) published work. I do not have further comments despite the few details noted below.

Given the important and strong foundation such results provide for further studies investigating *Toxoplasma gondii*'s impacts on host behavior and fitness in the wild (as well as its potential cascading effects at higher ecological levels), I would recommend publication in a broad audience journal such as Nature Communications.

Details:

- L74 I may rather add this reference to the Festschrift volume as a note (e.g., asterisk after “felids”) at the bottom of the page to not break the flow of the existing paragraph
- L81-83: either 1. 2. 3. or 1) 2) 3)
- L110 “Neither” or L111 “not” – one of these two terms has to be removed to not have double negation.
- Table 1 note c: sessed > assessed

Point-by-point response to reviewer comments

Reviewer #2

1. **Comment:** General comment: The authors did an excellent job of thoroughly and clearly responding to and addressing my (many) comments! Thanks for the additional thoughts and information, I learned a lot. I am satisfied with the changes in the MS. Please see below for a few minor comments/edits.

Response: We are pleased to have addressed major comments/concerns, and have responded to the remaining minor comments below.

2. **Comment:** Regarding the animals with a 'doubtful diagnosis', rather than excluding them, two sets of analyses could be run: 1) with them categorized within the 'positive' group; 2) with them categorized within the 'negative' group.

Response: We show below the results of the additional analyses among subadults and adults, which do not change the conclusions of the paper (*NB*, for cubs there were no 'doubtful' diagnoses).

In an adjusted subadult and adult model where 'doubtful' *T. gondii* diagnoses were included as 'infected', we found no difference in hyena approach distance from lions (-0.03 [95% CI: -0.92, 0.86]). The null findings match previous models.

In text, we report the results in which 'doubtful' diagnoses are included as negative (which is convention), and in sensitivity analyses we report results when 'doubtful' diagnoses are excluded.

3. **Comment:** Figure S1. SP Ration.... – should be ratio and missing end parenthesis. Also, please define SP in this caption. It should stand alone without the need to reference the MS. Also, suggest indicating that distance units are in meters. Figure S3 SPratio – should be SP ratio. Also, I would recommend staying consistent with color schemes, lack of consistency may cause considerable confusion or misinterpretation in a reader who scans or quickly reads the MS. In some figures red = positive, in others red = negative.

Response: Thank you for catching the typos. We have rectified the errors and also modified the color scheme for consistency throughout the manuscript.

4. **Comment:** Table S2 “.. they might swap effects...” should this be “SWAMP effects”?

Response: Thanks for catching this typo. We have corrected it.

Reviewer #3

1. **Comment:** I thank the authors for addressing my comments and questions regarding their manuscript - as well as my concern regarding their previous (non peer-reviewed) published work. I do not have further comments despite the few details noted below. Given the important and strong foundation such results provide for further studies investigating *Toxoplasma gondii*'s impacts on host behavior and fitness in the wild (as well as its potential cascading effects at higher ecological levels), I would recommend publication in a broad audience journal such as Nature Communications.

Response: We are grateful for the reviewer's feedback, which has undoubtedly improved the paper. Below, we've addressed the remaining comments.

2. **Comment:** L74 I may rather add this reference to the Festschrift volume as a note (e.g., asterisk after "felids") at the bottom of the page to not break the flow of the existing paragraph.

Response: Done.

3. **Comment:** L81-83: either 1. 2. 3. or 1) 2) 3)

Response: Done.

4. **Comment:** L110 "Neither" or L111 "not" – one of these two terms has to be removed to not have double negation.

Response: Done.

5. **Comment:** Table 1 note c: sessed > assessed

Response: Done